# ADAPTIVE RATIONAL ACTIVATIONS TO BOOST DEEP REINFORCEMENT LEARNING

**Quentin Delfosse**[*,1]**, Patrick Schramowski**[1,2,3]**, Martin Mundt**[1,3]**,**
**Alejandro Molina**[1] **& Kristian Kersting**[1,2,3,4]

[1] Computer Science Dept., TU Darmstadt        [2] German Center for Artificial Intelligence
[3] Hessian Center for Artifical Intelligence    [4] Centre for Cognitive Science, Darmstadt

## ABSTRACT

Latest insights from biology show that intelligence not only emerges from the connections between neurons, but that individual neurons shoulder more computational responsibility than previously anticipated. Specifically, neural plasticity should be critical in the context of constantly changing reinforcement learning (RL) environments, yet current approaches still primarily employ static activation functions. In this work, we motivate the use of adaptable activation functions in RL and show that rational activation functions are particularly suitable for augmenting plasticity. Inspired by residual networks, we derive a condition under which rational units are closed under residual connections and formulate a naturally regularised version. The proposed joint-rational activation allows for desirable degrees of flexibility, yet regularises plasticity to an extent that avoids overfitting by leveraging a mutual set of activation function parameters across layers. We demonstrate that equipping popular algorithms with (joint) rational activations leads to consistent improvements on different games from the Atari Learning Environment benchmark, notably making DQN competitive to DDQN and Rainbow.[1]

## 1 INTRODUCTION

Neural Networks' efficiency in approximating any function has made them the default choice in many machine learning tasks. This is no different in deep reinforcement learning (RL), where the DQN algorithm's introduction (Mnih et al., 2015) has sparked the development of various neural solutions. In concurrence with former neuroscientific explanations of brainpower residing in combinations stemming from trillions of connections (Garlick, 2002), present advances have emphasised the role of the neural architecture (Liu et al., 2018; Xie et al., 2019). As such, RL improvements have first been mainly obtained through enhancing algorithms (Mnih et al., 2016; Haarnoja et al., 2018; Banerjee et al., 2021) and only recently by searching for performing architectural patterns, via automatic deep policy search (Pang et al., 2021; Krishnan et al., 2023), or via decoupling object detection (Lin et al., 2020; Delfosse et al., 2023b) and policy search (Delfosse et al., 2023a; Wu et al., 2024).

However, research has also progressively shown that individual neurons shoulder more complexity than initially expected, with the latest results demonstrating that dendritic compartments can compute complex functions (*e.g.* XOR) (Gidon et al., 2020), previously categorised as unsolvable by single-neuron systems. This finding seems to have renewed interest in activation functions (Georgescu et al., 2020; Misra, 2020). In fact, many functions have been adopted across different domains (Redmon et al., 2016; Brown et al., 2020; Schulman et al., 2017). To reduce the bias introduced by a fixed activation function and achieve higher expressive power, one can further learn which activation function is performant for a particular task (Zoph & Le, 2017; Liu et al., 2018), learn to combine arbitrary families of activation functions (Manessi & Rozza, 2018), or find coefficients for polynomial activations as weights to be optimised (Goyal et al., 2019).

Whereas these prior approaches have all contributed to their respective investigated scenarios, there exists a finer approach that elegantly encapsulates the challenges brought on by reinforcement

---

[1]Rational library: github.com/k4ntz/activation-functions; Experiments: github.com/ml-research/rational_rl.

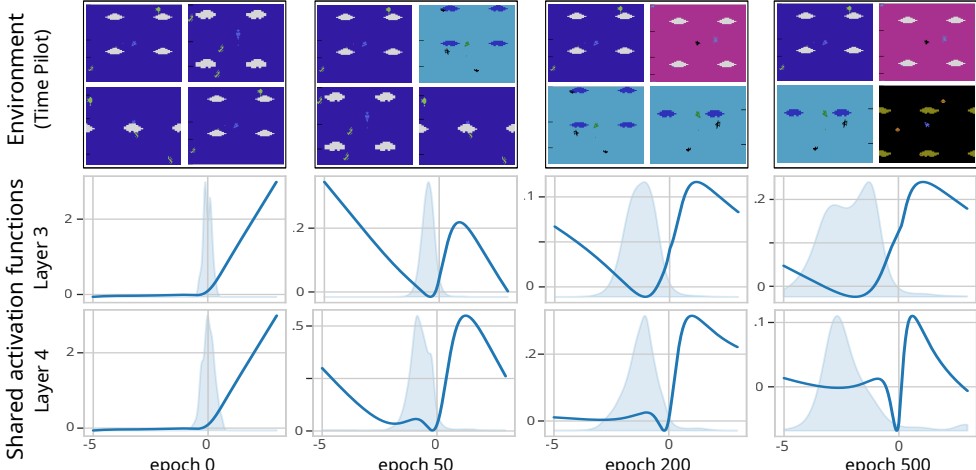

Figure 1: **Neural plasticity due to trainable activation functions allows RL agents to adapt to environments of increasing complexity.** Rational activations (bottom), with shared parameters in each of the last two layers, evolve together with their input distributions (shaded blue) when learning with DQN on Time Pilot. Each column corresponds to a training state where a new, more challenging part of the environment (top, e.g. increasing enemy speed and movement complexity) has been uncovered and is additionally used for training.

learning problems. Specifically, at each layer, we can learn rational activation functions (ratio of polynomials) (Molina et al., 2020). Not only can rationals converge to any continuous function, but they have further been proven to be better approximants than polynomials in terms of convergence (Telgarsky, 2017). Even more crucially, their ability to adapt while learning equips a model with high *neural plasticity* ( "capability to adjust to the environment and its transformations" (Garlick, 2002)). We argue that adapting to environmental changes is essential, making rational activation functions particularly suitable for dynamic RL environments. To provide a visual intuition, we showcase an exemplary evolution of two rational activation functions together with their respective changing input distributions in the dynamic "Time Pilot" environment in Fig. 1.

In this work, we **show that plasticity is of major importance for RL agents**, as a central element to satisfy the requirements originating from diverse and dynamic environments and **propose the use of rational activation functions to augment deep RL agents plasticity**. Apart from demonstrating the suitability of adaptive activation functions for Deep RL, we also evaluate how many additional layer weights can be replaced by rational activations. Our specific contributions are:

**(i)** We motivate why neural plasticity is a key aspect for Deep RL agents and that rational activations are adequate as adaptable activation functions. For this purpose, we not only highlight that rational activation functions adapt their parameters over time, but further prove that they can dynamically embed residual connections, which we refer to as residual plasticity.

**(ii)** As additional representational capacity can hinder generalisation, especially in RL (Farebrother et al., 2018; Roy et al., 2020; Yarats et al., 2021), we propose a joint-rational variant, that uses weight-sharing in rational activations across different layers.

**(iii)** We empirically demonstrate that rational activations bring significant improvements to DQN and Rainbow algorithms on Atari games and that our joint variant further increases performance.

**(iv)** Finally, we investigate the overestimation phenomenon of predicting too large return values, which has previously been argued to originate from an unsuitable representational capacity of the learning architecture (van Hasselt et al., 2016). As a result of our introduced (rational) neural and residual plasticity, such overestimation can practically be reduced.

We proceed as follows. We start off by arguing in favour of plasticity for deep RL, then show how rational functions are particularly suitable candidates to provide plasticity in neural networks and present our empirical evaluation. Before concluding, we touch upon related work.

Figure 2: **Neural plasticity is essential for reinforcement learning**. Human normalised mean scores for rigid (LReLU and CRELU) DQN agents, agents with non-rational, rational, tempered, and regularised plasticity are shown with standard deviation across 5 random seeded experimental repetitions. Larger scores are better. Tempered plasticity, allowing initial adaptation to the environments, but not their transformations in experimental repetitions, performs better on stationary environments. Regularised plasticity performs well across all environment types. Best viewed in colour. A description of the environments' types is provided in Appendix A.5.

## 2    RATIONAL PLASTICITY FOR DEEP RL

Let us start by arguing why deep reinforcement learning agents require extensive plasticity and show that parametric rational activation functions provide appropriate means to augment plasticity.

As motivated in the introduction, RL is subject to inherent distribution shifts. During training, agents progressively uncover new states (input drift) and, as the policy improves, the reward signal is modified (output drift). More precisely, for input drifts, we can distinguish environments according to how much they change through learning. For simplicity, we categorise according to three intuitive categories: stationary, dynamic and progressive environments. Consider the example of Atari 2600 games. Kangaroo and Tennis can be characterised as stationary since the games' distributions do not change significantly through learning. Asterix, Enduro and Q*bert are dynamic environments, as different distribution shifts (*e.g.* Cauldron, Helmet, Shield, etc. in Asterix) are provided to the agents in the first epochs, with no policy improvement required to uncover them. On the contrary, Jamesbond, Seaquest, and Time Pilot are progressive environments: agents need to master early stages before being provided with additional states, *i.e.* exposed to significant input shifts.

How do we efficiently improve RL agents' ability to adapt to environments and their changes? To deal with distribution shifts, our agents require high neural plasticity and thus benefit from adaptive architectures. To elaborate further in our work, let us consider the popular DQN algorithm (Mnih et al., 2015), that employs a $\boldsymbol{\theta}$-parameterised neural network to approximate the Q-value function of a state $S_t$ and action $a$. This network is updated following the Q-learning equation: $Q\left(S_t, a; \boldsymbol{\theta}\right) \equiv R_{t+1} + \gamma \max_a Q\left(S_{t+1}, a; \boldsymbol{\theta}\right)$. In addition to network connectivity playing an important role, we now highlight the importance of individual neurons by modifying the network architecture of the algorithm via the use of learnable activation functions, to show that they are a presently underestimated component. To emphasise the utility of the upcoming proposed rational and joint-rational activation functions, we will interleave early results into this section. The latter serves the primary purpose to not only motivate the suitability of the rational parameterisation to provide plasticity, but also discern the individual benefits of (joint-) rational activations, in the spirit of ablation studies.

### 2.1    RATIONAL NEURAL PLASTICITY

Rational functions are ratio of polynomials, defined on $\mathbb{R}$ by $\mathrm{R}(x) = \dfrac{\mathrm{P}(x)}{\mathrm{Q}(x)} = \dfrac{\sum_{j=0}^{m} a_j x^j}{1 + \sum_{k=1}^{n} b_k x^k}$,

where $x \in \mathbb{R}$, $\{a_j\}$ and $\{b_k\}$ are $m+1$ and $n$ (real) learnable parameters per layer. To test rational functions' plasticity, we use the discrete distribution shifts of the Permutted-MNIST continual learning experiment. We show in Appendix 4 that rational activation functions improve the plasticity over both ReLU and over CRELU (Shang et al., 2016), used to augment plasticity by Abbas et al. (2023). For RL we show in Fig. 2 that the rational parametrisations substantially enhances RL agents. More precisely, by comparing agents with rigid networks (a fixed Leaky ReLU baseline) to agents with rational plasticity (*i.e.* with a rational activation function at each layer), we see that rational functions boosts the agents to super-human performances on 7 out of 9 games. The acquired extra neural plasticity seems to play a significant role in these Atari environments, especially in progressive ones.

In order to discern the benefits of general plasticity through any adaptive activation function, over the proposed use of the rational parametrisation, Fig. 2 additionally includes agents with Concatenated ReLU and with Parametrised Exponential Linear Unit (Trottier et al., 2017). CReLU, was used in RL to address plasicity loss (Abbas et al., 2023), outperforms LRELU on 6 out of 9 games, but is always outmatched by rational plasticities. PELU that uses 3 parameters to control its slope, saturation and exponential decay, and has been shown to outperform other learnable alternatives on classification tasks (Godfrey, 2019). However, in contrast to the rational parameterisation, it seems to fall behind and only boosts the agents to super-human performance on 3 out of 9 games (contrary to 7), implying that the type of plasticity provided by rational activations is particularly suitable.

To highlight the desirability of rational activations even further, we additionally distinguish between the plasticity of agents towards their specific environment and the plasticity allowing them to adapt while the environment is changing. To this end, we show agents equipped with rational activations that are tempered in Fig. 2. Such agents are equipped with the final, optimised rational functions of trained rational-equipped agents. They correspond to frozen functions from agents that already adapted to their specific environment. The plasticity of the rationals is thus tempered (*i.e.* stopped) in a repeated application (*i.e.* another training session) to emphasise the necessity to continuously adapt the activation functions, together with the layers' weights during training. Whereas providing agents with such tempered, tailored to the task, activations already boosts performances, rational plasticity at all times is essential, particularly in dynamic and progressive environments.

## 2.2 RATIONAL RESIDUAL PLASTICITY

The prior paragraphs have showcased the advantage of agents equipped with rational activation functions, rooted in their ability to update their parameters over time. However, we argue that the observed boost in performance is not only due to parameters adapting to distributional drifts. Rational activations can embed one of the most popular techniques to stabilise deep networks training; namely, they can dynamically make use of a residual connection. We refer to this as residual plasticity.

**Rationals are closed under residual connection and provide residual plasticity.** We here show that rational functions with strictly higher degree in the numerator embed residual connections. Recall that residual neural networks (ResNets) were initially introduced following the intuition that it is easier to optimise the residual mapping than to optimise the original, unreferenced mapping (He et al., 2016). Formally, residual blocks of ResNets propagate an input $X$ through two paths: a transforming block of layers that preserves the dimensionality ($F$) and a residual connection (identity).

**Theorem:** Let R be a rational function of order (m, n). R embeds a residual connection $\Leftrightarrow m > n$.

*Proof:* Let us consider a rational function R = P/Q of order $(m, n)$, with coefficients $A^{[m]} = (a_j)_{j=0}^m \in \mathbb{R}^{m+1}$ of P and $B^{[n]} = (b_i)_{i=0}^n \in \mathbb{R}^{n+1}$ of Q (with $b_0 = 1$). We denote by $\otimes$ (respectively $\oslash$) the Hadamard product (respectively division). Let $X \in \mathbb{R}^{n_1 \times \cdots \times n_x}$ be a tensor corresponding to the input of the rational function of an arbitrary layer in a given neural network. We derive $X^{\otimes k} = \bigotimes_{i=1}^k X$. Furthermore, we use $GV^{[k]}(X) = [\mathbf{1}, X, X^{\otimes 2}, ..., X^{\otimes k}] \in \mathbb{R}^{(n_1 \times \cdots \times n_x) \times k+1}$ to denote the tensor containing the powers up to $k$ of the tensor $X$. Note that $GV^{[k]}$ can be understood as a generalised Vandermonde tensor, similar as introduced in (Xu et al., 2016). For $V^{[k]} = (v_i)_{i=0}^k \in \mathbb{R}^{k+1}$, let $GV^{[k]}.V^{[k]} = \sum_{i=0}^k v_i X^{\otimes i}$ be the weighted sum over the tensor elements of the last dimension.

Now, we apply the rational activation function R with residual connection to $X$:

$$\mathbf{y}(X) = \mathrm{R}(X) + X = GV^{[m]}(X).A^{[m]} \oslash GV^{[n]}(X).B^{[n]} + X$$
$$= (GV^{[m]}(X).A^{[m]} + X \otimes GV^{[n]}(X).B^{[n]}) \oslash GV^{[n]}(X).B^{[n]}$$
$$= (GV^{[m]}(X).A^{[m]} + GV^{[n+1]}(X).B_0^{[n+1]}) \oslash GV^{[n]}(X).B^{[n]}$$
$$= GV^{[\max(m,n+1)]}(X).C^{[\max(m,n+1)]} \oslash GV^{[n]}(X).B^{[n]} \qquad = \widetilde{\mathrm{R}}(X),$$

where $B_0^{[n+1]} = (b_{0,i})_{i=0}^{n+1} \in \mathbb{R}^{n+2}$ (with $b_{0,0} = 0$ and $b_{0,i} = b_{i-1}$ for $i \in \{1, ..., n+1\}$), $C^{[\max(m,n+1)]} = (c_j)_{j=0}^{\max(m,n+1)}$ ($c_j = a_j + b_{j-1}$, $a_j = 0 \ \forall j \notin \{0, ..., m\}$, $b_j = 0 \ \forall j \notin \{0, ..., n\}$).
$\widetilde{\mathrm{R}}$ is a rational function of order $(m', n')$, with $m' > n'$. In other words, rational activation functions of order $m > n$ embed residual connections. Using the same degrees for numerator and denominator

certifies asymptotic stability, but our derived configuration allows rationals to implicitly use residual connections. Importantly, note that these residual connections are not rigid, as these functions can progressively learn $a_j = 0$ for all $j > n$, *i.e.* we have residual plasticity.

**Rational plasticity to replace residual blocks' plasticity.** In very deep ResNets, it has been observed that feature re-combinations does not occur inside the blocks but that transitions to representations occur during dimensionality changes Veit et al. (2016); Greff et al. (2017).
To investigate this hypothesis, Veit et al. have conducted lesioning experiments, where a residual block is removed from the network, and surrounding ones are fine-tuned to recover. Whereas we emphasize that we do not claim that the residual in rationals can replace entire convolutional blocks or that they are generally equivalent, we hypothesize that under the conditions investigated by Veit et al. of very deep networks, residual blocks could learn complex activation function-like behaviours. To test this conjecture, we repeat the lesioning experiments, but also test replacing the lesioned block with a rational function that satisfies the residual connection condition derived above. Results are provided in appendix (*cf.* A.2) and show that **rational functions' plasticity can efficiently compensate for the lost capacity of lesioned residual blocks in very deep residual networks**.

### 2.3 NATURAL RATIONAL REGULARISATION

We have motivated and shown that the combination of neural and residual plasticity form the central pillars for why rational activation functions are desirable in deep RL. In particular for dynamic and progressive environments, rational plasticity has been observed to provide a substantial boost over alternatives. However, if we circle back to Fig. 2 and take a more careful look at the stationary environments, we can observe that our previously investigated tempered rational plasticity (for emphasis, initially allowed to tailor to the task but later "stopped" in experimental repetition) can also have an upper edge over full plasticity. The extra rational plasticity at all times might reduce generalisation abilities, particularly on non-diverse stationary environments. In fact, prior works have highlighted the necessity for regularisation methods in deep reinforcement learning (Farebrother et al., 2018; Roy et al., 2020; Yarats et al., 2021).

We thus propose a naturally regularised rational activation version, inspired from residual blocks. In particular, Greff et al. have indicated that sharing the weights can improve learning performances, as shown in Highway (Lu & Renals, 2016) and Residual Networks (Liao & Poggio, 2016). In the spirit of these findings, we propose the regularised *joint-rationals*, where we constrain the input to propagate through different layers but always be activated by the same learnable rational activation function. Rational functions thus share a mutual set of parameters across the network, (instead of layers, *cf.* Fig. 11). As observable in Fig. 2, this regularised form of plasticity increases the agents' scores in the stationary environments and does not deteriorate performances in the progressive ones.

### 3 EMPIRICAL EVIDENCE FOR PLASTICITY

Our intention here is to investigate the benefits of neural plasticity through rational networks for deep reinforcement learning. That is, we investigated the following questions:

**(Q1)** Do neural networks equipped with rational plasticity outperform rigid baselines?

**(Q2)** Can neural plasticity make up for more heavy algorithmic RL advancements?

**(Q3)** Can plasticity address the overestimation problem?

**(Q4)** How many more parameters would rigid networks need to measure up to rational ones?

To this end, we compare[2] our rational plasticity using the original DQN algorithm and its convolutional network (Mnih et al., 2015) on 15 different games of the Atari 2600 domain (Brockman et al., 2017). We compare these architectures to ones equipped with Leaky ReLU (as experiments on Breakout and SpaceInvaders showed that agents with Leaky ReLU outperform ReLU ones), the learnable PELU function, as well as SiLU ($\text{SiLU}(x) = x \cdot \text{sigmoid}(x)$) and its derivative dSiLU. Elfwing et al. showed that SiLU or a combination of SiLU (on convolutional layers) and its derivative (on fully connected layers) perform better than ReLU in DQN agents on several games (2018). SiLU and dSiLU are —to our knowledge— the only activation functions specifically designed for RL applications. More details on the architecture and hyperparameters can be found in Appendix A.8.

---

[2]30.000 GPU hours, carried out on a DGX-2 Machine with Nvidia Tesla V100 with 32GB.

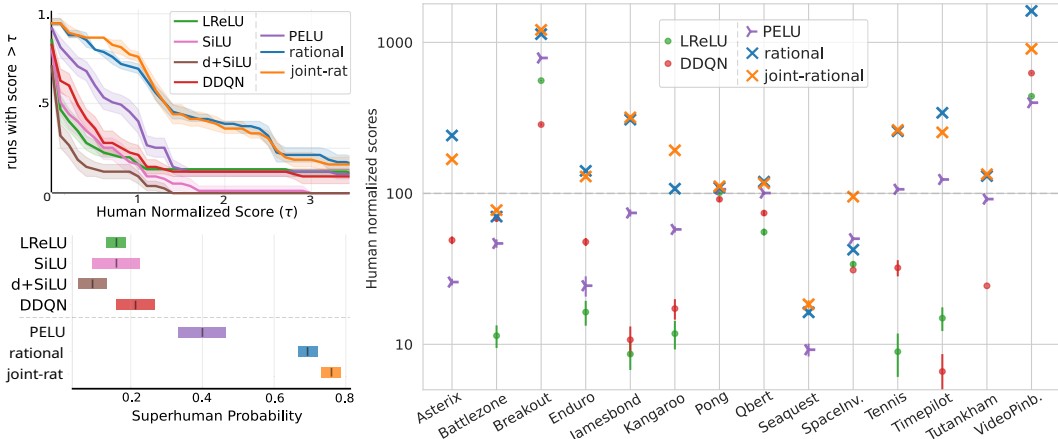

Figure 3: **Learnable functions' plasticity boosts RL agents.** For reliable evaluation, we report the performance profiles (top left) as well as superhuman probabilities (with CIs, bottom left) of baselines (*i.e.* DQN and DDQN with Leaky ReLU, DQN with SiLU and SiLU + dSiLU), as well as DQN with plasticity: using PELU, rational and joint-rational (5 random seeds). While the learnable PELU already augment performances of its agents, rational and joint-rational ones lift them above human performances on more than 70% of our runs. Detailed score tables are provided in Appendix A.3.

We then compare increased neural plasticity provided by (joint-)rational networks to algorithm improvements, namely the Double DQN (DDQN) method (van Hasselt et al., 2016), that tackles DQN's overestimation problem, as well as Rainbow (Hessel et al., 2018). Rainbow incorporates multiple algorithm improvements brought to DQN—Double Q-learning, prioritised experience replay, duelling network, multi-step target, distributional learning and stochastic networks—and is widely used also as a baseline (Lin et al., 2020; Hafner et al., 2021). We further explain how neural plasticity can help readdress overestimation. Finally, we evaluate how many additional weights rigid networks need to approximate rational ones.

In practice, we used safe rational functions (Molina et al., 2020), *i.e.* we used the absolute value of the sum in the denominator to avoid poles. This stabilises training and makes the function continuous. Rational activation functions are shared across layers (adding only 10 parameters per layer) or through the whole network for the regularised (joint) version, with their parameters optimised together with the rest of the weights. For ease of comparison and reproducibility, we conducted the original DQN experiment (also used by DDQN and SiLU authors) using the mushroomRL (D'Eramo et al., 2020) library, with the same hyperparameters (*cf.* Appendix A.8) across all the Atari agents, for a specific game, but we did not use reward clipping. For a fair comparison, we report final performances using the human-normalised (*cf.* Eq. 2 in Appendix) mean and standard deviation of the scores obtained by fully trained agents over five seeded reruns for every (D)DQN agent. However, since often only the best performing RL agent is reported in the literature, we also provide tables of such scores (*cf.* Appendix A.3). For the Rainbow algorithm, we unfortunately can only report the results of single runs. A single run took more than 40 days on an NVIDIA Tesla V100 GPU; Rainbow is computationally quite demanding (Obando-Ceron & Castro, 2021).

**(Q1) DQN with activation plasticity is better than rigid baselines.**    To start off, we compared RL agents with additional plasticity (from PELU and rationals) to rigid DQN baselines: Leaky ReLU, as well as agents equipped with SiLU and SiLU+dSiLU activation functions.
The results summarised in Fig. 3 confirm what our earlier figure had shown, but on a larger scale. While RL agents with functions of the SiLU family do not outperform Leaky ReLU ones in our games, plastic DQN agents clearly outperform their rigid activation counterparts. DQN with regularised plasticity even obtains a higher superhuman probability and highest mean scores 64% of the time. Scores on (difficult credit assignment) Skiing are in Appendix A.3. This clearly shows that plasticity, and above all rational plasticity, pays off for deep agents, providing an affirmative answer to **Q1**.

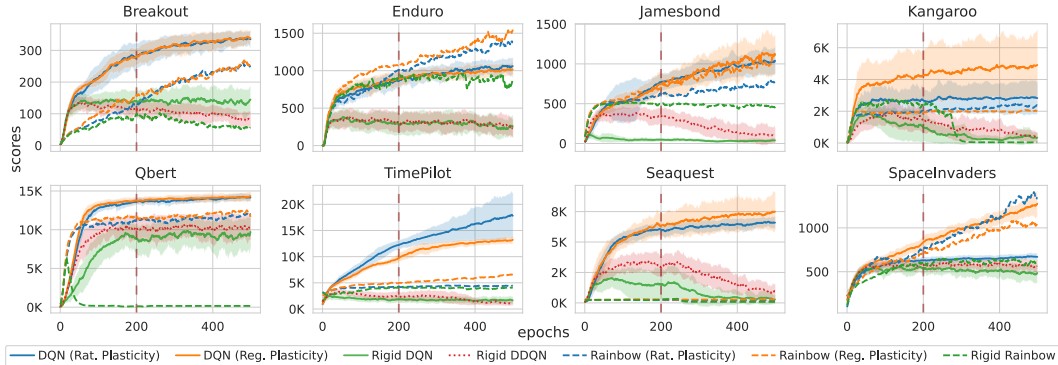

Figure 4: Networks with rational (Rat.) and regularised (Reg.) rational plasticity compared to rigid baselines (DQN, DDQN and Rainbow) over five random seeded runs on eight Atari 2600 games. The resulting mean scores (lines) and standard deviation (transparent area) during training are shown. As one can see, DDQN does not resolve performance drops but only delays them (e.g. particularly pronounced on Seaquest). A figure including the evolution of every agent on all Atari 2600 games is provided in Appendix A.4. Figure best viewed in colour.

**(Q2) Neural plasticity can boost performances of complex deep reinforcement learning approaches, such as Rainbow.** In Fig. 4, we show the learning curves of Rainbow and DQN agents, both with Leaky ReLU baselines, as well as with full and regularised rational plasticity types. While Rainbow is computationally much heavier ($\sim 8$ times slower than DQN in our experiments, with higher memory needs), its rigid form never outperforms the much simpler and more efficient DQN with neural plasticity, and its rational versions dominate in only 1 out of 8 games (Enduro). In our experiments, Rainbow even lost to vanilla DQN on 3 games. These results show that augmenting the plasticity of an RL agent's modeling architecture can be of higher importance than bringing complex and computationally expensive improvements to the learning algorithm.

Therefore, DQN with rational plasticity is a competitive alternative to the complicated and expensive Rainbow method. Plasticity also improves Rainbow agents, answering question **(Q2)** affirmatively.

**(Q3) Neural plasticity directly tackles the overestimation problem.** Revisiting Fig. 4, one can see that Rainbow variants are worst on dynamic environments such as Jamesbond, Time Pilot and particularly Seaquest. For these games, the performance of rigid (Leaky ReLU) DQN progressively decreases. Such drops are well known in the literature and are typically attributed to the overestimation problem of DQN. This overestimation is due to the combination of bootstrapping, off-policy learning and a function approximator (neural network) operating by DQN. van Hasselt et al. showed that inadequate flexibility of the function approximator (either insufficient or excessive) can lead to overestimations of a state-action pairs (2016). The max operator in the update rule of DQN then propagates this overestimation while learning with the replay buffer. The overestimated states can stay in the buffer long before the agent revisit (and thus update) them. This can lead to catastrophic performance drops. To mitigate this problem, van Hasselt et al. introduced a second network to separate action selection from action evaluation, resulting in Double DQN (DDQN).

We have compared rigid DDQN (equipped with Leaky ReLU), to vanilla DQN with neural plasticity on Atari games. As one can see in Fig. 3, DQN with rational plasticity outperforms the more complex (rigid) DDQN algorithm on every considered Atari game. This reinforces the affirmative answer to **(Q1)** from earlier on. More importantly, we have computed the relative overestimation values of the (D)DQN, both with and without neural plasticity, following: overestimation $= \frac{\text{Q-value} - R}{R}$, where the return $R$ corresponds to $R = \sum_{t=0}^{\infty} \gamma^t r_t$, with the observed reward $r_t$ and the discount factor $\gamma$.

The results are summarised in Fig. 5. Plasticity helps to reduce overestimation drastically. DDQN substantially reduces overestimation on Jamesbond, Kangaroo, Tennis, Time Pilot and Seaquest. For these games, DDQN obtains the best performances among all rigid variants only on Jamesbond (*cf.* Tab. 3 in Appendix). Moreover, Fig. 4 reveals that the DDQN agents' performance drops are only delayed and not prevented. The performance drops thus happen after the 200th epoch, after which the agents' training is usually stopped, as no more performance increase seems achievable.

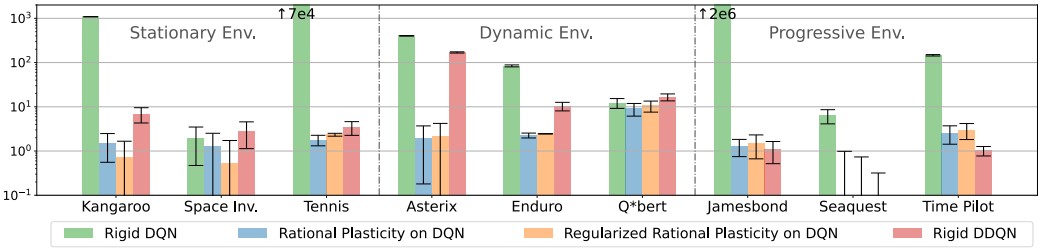

Figure 5: **Plasticity naturally reduces overestimation.** Relative overestimation values ($\downarrow$, log scale) of rigid DQN and DDQN, as well as DQN with rational and regularised rational plasticity. Each trained agent is evaluated on 100 completed games (5 seeds per game per agent). Agents with rational plasticity lower overestimation values as much or further than rigid DDQN ones, which has specifically been introduced to this end. Figure best viewed in colour.

Overestimation might play a role in the performance drops on progressive environments (*cf.* Fig. 4: Jamesbond, TimePilot and Seaquest), but cannot fully explain the phenomena. RL agents with higher plasticity handle these games much better while embedding only a few more parameters. Hence, we advocate that neural plasticity better deals with distribution shifts of dynamic and progressive environments. Perhaps surprisingly, (regularised) rational plasticity not only works well on challenging progressive environments but also on simpler ones such as Enduro, Pong and Q*bert, where more flexibility is likely to hurt. Flexibility does not lead to overestimation (*cf.* Fig. 5). The rational functions for these games have a simpler profile than ones of more complicated games like Kangaroo and Time Pilot (*cf.* Appendix A.6). The rational functions seem to adapt to the environment's complexity and the policy they need to model. This clearly provides an affirmative answer to **(Q3)**.

**(Q4) Adding parameters through rationals efficiently augments plasticity.** Compared to rigid alternatives, the joint-rational networks embed in total 10 additional parameters and always outperform (*cf.* Fig. 4) PELU (12 more parameters) ones. Our proposed method to add plasticity via rational functions thus efficiently augments the capacity of the network. However, ReLU layers can theoretically approximate rational functions (Telgarsky, 2017). Augmenting the number of layers (or neurons per layer) is thus, theoretically, a costly alternative to augment the plasticity. How many parameters are practically needed in rigid networks to obtain similar performances? Searching for bigger equivalent architectures for RL agents is tedious, as RL training curves possess considerably more variance and noise than SL ones (Pang et al., 2021), but this question is not restricted to RL. We thus answer it by highlighting the generality of our insights, demonstrated by further investigation on a classification scenario (*cf.* Appendix A.1). In short, rigid baselines need up to 3.5 times as many parameters as the architectures that use rational functions in order to obtain similar performances.

All experimental results together clearly show that increasing neural plasticity, particularly through the integration of rational activations functions, considerably benefits deep reinforcement learning agents in a highly computational efficient manner.

## 4 RELATED WORK

Next to the related work discussed throughout the paper, our work on neural plasticity is also related to several research lines on neural architecture search and to the choice of activation functions, particularly in deep reinforcement learning settings.

**The choice of activation functions.** Many functions have been adopted across domains (*e.g.* Leaky ReLU in YOLO (Redmon et al., 2016), hyperbolic tangent in PPO (Schulman et al., 2017), GELU in GPT-3 (Brown et al., 2020)), indicating that the relationship between the choice of activation functions and the performances is highly dependent on the task, architecture and hyper-parameters. As shown, parametric functions augment on plasticity. Molina et al. showed that rational functions can outperform other learnable activation function types on supervised learning tasks (2020). Telgarsky (2017) showed that rationals are locally better approximants than polynomials. Loni et al. (2023) showed that searching for activation functions mitigates the performance drops of sparsity in networks.

**Neural Architectures for Deep Reinforcement Learning.** Cobbe et al. showed that the architecture of IMPALA (Espeholt et al., 2018), notably containing residual blocks, improved the performances over the original Nature-CNN network used in DQN (2019). Motivated by these findings, Pang et al. (2021) recently applied neural architecture search to RL tasks and demonstrated that the optimal architecture highly depend on the environment. Their search provides different architectures across environments, with varying activation functions across layers and potential residual connections. Continuously modifying the complexity of the neural network based on the noisy reward signal in a complex architectural space is extremely resource demanding, particularly for large scale problems. Many reinforcement learning specific problems, such as noisy rewards (Henderson et al., 2018), input interdependency (Mnih et al., 2015), policy instabilities (Haarnoja et al., 2018), sparse rewards, difficult credit assignment (Mesnard et al., 2021), complicate an automated architecture search.

**Plasticity in deep RL.** A lot of attention has recently been brought to the plasticity of RL agents' learning structures. Abbas et al. (2023) have also identified their loss of plasticity and answered it using concatenated ReLU (CReLU) in Rainbow. Nikishin et al. (2022) periodically reset parts of the networks, Sokar et al. (2023) improved the resets by targeting identified dormant neurons. Similarly, Nikishin et al. (2023) inject plasticity via incorporating new trainable weights. Lyle et al. (2022) mitigate capacity (or plasticity) loss, regularizing some features back to their starting values, and later showed that layer normalization help with plasticity (Lyle et al., 2023). Dohare et al. (2023) tackle dynamics with continual backprop and apply it to RL on PPO (Dohare et al., 2021), also varying between different non-learnable activation functions. Dynamically adapting the hyperparameter landscape is also improves agents' adaptability to distribution shifts (Zahavy et al., 2020; Mohan et al., 2023). Testing how much much of these techniques can be covered by the use of rational plasticity is an interesting line of future work, as rational functions dynamically change the weights optimisation landscape. Fuks et al. (2019) adjust sub-policies on sub-games to find suitable hyperparameters that bootstrap a main evolution-based optimised agent. Apart from using CReLU, all of these techniques are complementary to the use of rational plasticity.

## 5 LIMITATIONS, FUTURE WORK AND SOCIETAL IMPACT

We have shown the benefits of rational activation functions for RL, as a consequence of both their neural and residual plasticity. In our derivation for closure under residuals, we have deduced that the degree of the polynomial in the numerator needs to be greater than that of the denominator. Correspondingly, we have based our empirical investigations on the degrees $(5, 4)$. Interesting future work would be to further automatically select suitable such degrees, or even integrating rationals into dynamic hyperparameters' optimisation techniques. One should also explore neural plasticity in more advanced RL approaches, including short term memory (Kapturowski et al., 2019), neurosymbolic approaches (Delfosse et al., 2024), finer exploration strategy (Badia et al., 2020), and in continual learning (Kudithipudi et al., 2022) techniques. Finally, the noisy optimisation performed in our RL experiments contribute to carbon emissions. However, this is usually a means to an end, as RL algorithms are also used to optimise energy distribution and consumption in several applications.

## 6 CONCLUSION

In this work, we have highlighted the central role of neural plasticity in deep reinforcement learning algorithms, and have motivated the use of rational activation functions, as a lightweight way of boosting RL agents performances. We derived a condition, under which rational functions embed residual connections. Then the naturally regularised joint-rational activation function was developed, inspired by weight sharing in residual networks.

The simple DQN algorithm equipped with these (regularised) rational forms of plasticity becomes a competitive alternative to more complicated and costly algorithms, such as Double DQN and Rainbow. Fortunately, the complexity of these rational functions also seem to automatically adapt to the one of the environment used for training. Their use could be a substitute for more expensive architectural searches. We thus hope that they will be adopted in future deep reinforcement learning algorithms, as they can provide agents with the necessary neural plasticity required by stationary, dynamic and progressive reinforcement learning environments.

ACKNOWLEDGEMENTS

The authors thank Elisa Corbean for her help on the manuscript revisions, as well as the anonymous reviewers of ICLR 2024 for their valuable feedback. This research work has been funded by the German Federal Ministry of Education and Research and the Hessian Ministry of Higher Education, Research, Science within their joint support of the National Research Center for Applied Cybersecurity ATHENE, via the "SenPai: XReLeaS" project, from the German Center for Artificial Intelligence (DFKI). This work was also supported by the project "safeFBDC - Financial Big Data Cluster (FKZ: 01MK21002K)", funded by the German Federal Ministry for Economics Affairs and Energy as part of the GAIA-x initiative. It benefited from the Hessian Ministry of Higher Education, Research, Science and the Arts (HMWK; project "The Third Wave of AI")

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

# A   APPENDIX

As mentioned in the main body, the appendix contains additional materials and supporting information for the following aspects: rational activation functions improving plasticity (4), comparison of rational and rigid networks with different sizes on supervised learning experiments (A.1), results on replacing residual blocks with rational activation functions (A.2), every final and maximal scores obtained by the reinforcement learning agents used in our experiments (A.3), the evolutions of these scores (A.4), the different environment types with illustrations of their changes (A.5), graphs of the learned rational activation functions (A.6) and technical details for reproducibility (A.8).

Rational functions improve plasticity

To prove that rational can help with plasiticy, we tested them in continual learning settings (with more abrupt distribution shifts). We included Concatenated RELU and rational functions to an existing implementation of continual AI[3], in which 4 layers (2 convolutional ones and to fully connected ones) networks are trained on MNIST. The network then continues training on PERM.1, a variation of the dataset, for which a fixed random permutation is applied to every image. Another permutation is used for PERM. 2, used after the training on PERM1. As shown in Fig. 6, networks with rationals are both better at modelling the new data (higher accuracies on the currently trained data), but are also able to retain more information about the data previously trained on. Networks with Continual ReLU (Shang et al., 2016) better retain information on Task 1, while performing on par with ReLU ones for the 2 other tasks.

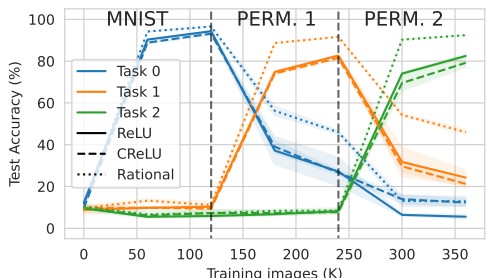

Figure 6: **Rational function improve plasticity on the permutted MNIST experiment.** Rational networks obtain better accuracies on each currently and previously trained datasets.

## A.1   RATIONAL EFFICIENT PLASTICITY CAN REPLACE LAYER'S WEIGHT PLASTICITY

We here show that networks with rational activations not only outperform Leaky ReLU ones with the same amount of parameters, but also to outperform deeper and more heavily parametrised neural networks (indicated by the colours). For example, a rational activated VGG4 not only performs better than a rigid Leaky ReLU VGG4 at 1.37M parameters, but even performs similarly to the 4.71M parameters rigid VGG6. Activation's plasticity allowing to reduce the number of layers weights is also shown by the experiments summarized in Tab. 2 in the next section, where blocks from a pretrained ResNet are replaced by a rational function, and the resulting networks are able to recover and surpass their accuracies.

| Architecture | | VGG4 | | VGG6 | | VGG8 | |
|---|---|---|---|---|---|---|---|
| Activation function | | LReLU | Rational | LReLU | Rational | LReLU | Rational |
| CIFAR 10 | Training Acc@1 | 83.0±.3 | 87.1±.6 | 86.9±.2 | 89.2±.2 | 90.1±.1 | 92.4±.2 |
| | Testing Acc@1 | 80.0±1. | 84.3±.5 | 83.1±.6 | 85.4±.6 | 85.0±1. | 86.9±.3 |
| CIFAR 100 | Training Acc@1 | 64.6±.8 | 70.4±.9 | 70.7±.6 | 86.0±.9 | 87.7±.2 | 87.8±.1 |
| | Testing Acc@1 | 56.5±.9 | 58.9±.6 | 59.0±.5 | 59.9±.9 | 60.0±.9 | 59.9±.4 |
| # Network parameters | | 1.37M | | 4.71M | | 9.27M | |

Table 1: Shallow rational networks perform as deeper Leaky ReLU ones. VGG networks training and testing top-1 accuracies with different numbers of layers are evaluated on CIFAR10 and CIFAR100. Rational VGG4 has similar performances as VGG6 network, with 3.5 times less parameters, and Rational VGG6 outperforms VGG8, with two times less parameters. Shaded colour pairs included for emphasis.

---

[3]https://github.com/ContinualAI/colab/blob/master/notebooks/permuted_and_split_mnist.ipynb

## A.2 Residual block learn of deep ResNet learn activation function-like behaviour.

We present in this section lesioning experiments, where a residual block is lesioned from a pretrained Residual Network, and the surrounding blocks are fine-tuned (with a learning rate of 0.001) for 15 epochs. These lesioning experiments were first conducted by Veit et al. (2016). We also perform rational lesioning, where we replace a block by an (identity initialised)[4] rational activation function (instead of removing the block), and train the activation function along with the surrounding blocks. The used rational functions have the same order as in every other experiment ($(m, n) = (5, 4)$), that satisfies the rational residual property derive in the paper). We report recovery percentages, computed following:

$$\text{recovery} = 100 \times \frac{\text{finetuned} - \text{surgered}}{\text{original} - \text{surgered}}. \tag{1}$$

We also provide the amount of dropped parameters of each lesioning.

Table 2: Rational functions improve lesioning. The recovery percentages for finetuned networks after lesioning (Veit et al., 2016) of a ResNet layer's (L) block (B) are shown. Residual blocks were lesioned, *i.e.* replaced with the identity (Base) or a rational from a pretrained ResNet101 (44M parameters). Then, the surrounding blocks (and implanted rational activation function) are retrained for 15 epochs. Larger percentages are better, best results are in **bold**.

| Recovery (%) | Lesioning | L2B3 | L3B19 | L3B22 | L4B2 |
|---|---|---|---|---|---|
| Training | Original (Veit et al., 2016) | 100.9 | 90.5 | 100 | 58.9 |
| | Rational (ours) | **101.1** | **104** | **120** | **91.1** |
| Testing | Original (Veit et al., 2016) | **93.1** | 97.1 | 81.6 | 81.7 |
| | Rational (ours) | 90.5 | **97.6** | **91.5** | **85.3** |
| | % dropped params | 0.63 | 2.51 | 2.51 | 10.0 |

As the goal is to show that flexible rational functions can achieve similar modelling capacities to the residual blocks, we did not apply regularisation methods and mainly focused on training accuracies. We can clearly observe that rational activation functions lead to performance improvements that even surpass the original model, or are able to maintain performances when the amount of dropped parameters rises.

## A.3 Complete scores table for Deep Reinforcement Learning

Through this work, we showed the performance superiority of reinforcement learning agents that embed additional plasticity provided by learnable rational activation functions. We used human normalised scores (*cf.* Eq. 2) for readability. For completeness, we provide in this section the final raw scores of every trained agent. As many papers provide the maximum obtained score among every epoch and every agent, even if we consider it to be an inaccurate and noisy indicator of the performances, for which random actions can still be taken (because of $\epsilon$-greedy strategy also being used in evaluation). A fairer indicator to compare methods is the mean score. We thus also provide final mean scores (of agents retrained among 5 seeded reruns) with standard deviation. We start off by providing the human scores used for normalisation (provided by van Hasselt et al., in Table 5), then provide final mean and maximum obtained raw scores of every agent.

---

[4]all weights are initially set to 0 but $a_1$ (and $b_0$), both set to 1.

| Algorithm | DQN | | | DDQN | | DQN with Plasticity | |
|---|---|---|---|---|---|---|---|
| Activation | LReLU | SiLU | d+SiLU | LReLU | PELU | rational | joint-rational |
| Asterix | $1.85_{\pm1.2}$ | $0.52_{\pm0.6}$ | $2.14_{\pm1.4}$ | $48.9_{\pm17.7}$ | $25.8_{\pm3.7}$ | $\mathbf{242}_{\pm23.5}$ | $168_{\pm32.6}\bullet$ |
| Battlezone | $11.4_{\pm7.0}$ | $21.2_{\pm15.0}$ | $11.3_{\pm6.7}$ | $68.2_{\pm34.8}$ | $46.6_{\pm19.5}$ | $70.1_{\pm2.1}\bullet$ | $\mathbf{77.4}_{\pm8.7}$ |
| Breakout | $558_{\pm166}$ | $93.9_{\pm57.6}$ | $11.7_{\pm14.0}$ | $286_{\pm122}$ | $788_{\pm79.2}$ | $1134_{\pm130}\bullet$ | $\mathbf{1210}_{\pm36.0}$ |
| Enduro | $16.3_{\pm21.3}$ | $37.0_{\pm17.7}$ | $0.37_{\pm0.5}$ | $47.7_{\pm18.1}$ | $24.5_{\pm42.6}$ | $\mathbf{141}_{\pm15.0}$ | $129_{\pm14.7}\bullet$ |
| Jamesbond | $8.62_{\pm6.4}$ | $6.08_{\pm3.7}$ | $5.28_{\pm4.4}$ | $10.7_{\pm11.1}$ | $74.2_{\pm51.5}$ | $308_{\pm48.5}\bullet$ | $\mathbf{312}_{\pm59.5}$ |
| Kangaroo | $11.8_{\pm12.5}$ | $128_{\pm95.6}\bullet$ | $13.9_{\pm18.5}$ | $17.2_{\pm14.5}$ | $57.7_{\pm14.6}$ | $107_{\pm43.1}$ | $\mathbf{193}_{\pm86.8}$ |
| Pong | $101_{\pm5.5}$ | $96.1_{\pm12.0}$ | $104_{\pm3.3}$ | $91.3_{\pm30.8}$ | $106.4_{\pm2.2}$ | $107.0_{\pm2.4}\bullet$ | $\mathbf{107.3}_{\pm2.7}$ |
| Qbert | $55.4_{\pm17.1}$ | $14.2_{\pm17.0}$ | $2.74_{\pm0.2}$ | $74.0_{\pm21.7}$ | $101_{\pm6.6}$ | $\mathbf{120}_{\pm2.8}$ | $117_{\pm4.9}\bullet$ |
| Seaquest | $0.57_{\pm0.4}$ | $3.67_{\pm4.1}$ | $0.18_{\pm0.2}$ | $2.17_{\pm0.9}$ | $9.21_{\pm2.5}$ | $16.3_{\pm0.5}\bullet$ | $\mathbf{18.4}_{\pm3.3}$ |
| Skiing | $-90.7_{\pm37.9}$ | $-111_{\pm-0.7}$ | $-85.5_{\pm43.4}$ | $-86.9_{\pm46.6}$ | $-111_{\pm-7}$ | $\mathbf{-59.5}_{\pm60.7}$ | $-60.2_{\pm56.1}\bullet$ |
| Space Inv. | $33.9_{\pm4.3}$ | $33.1_{\pm11.9}$ | $32.4_{\pm12.4}$ | $31.0_{\pm1.0}$ | $50.1_{\pm3.3}\bullet$ | $42.3_{\pm3.1}$ | $\mathbf{95.1}_{\pm17.7}$ |
| Tennis | $8.94_{\pm17.3}$ | $26.3_{\pm53.3}$ | $78.5_{\pm64.3}$ | $32.1_{\pm51.6}$ | $106_{\pm53.3}$ | $257.8_{\pm2.8}\bullet$ | $\mathbf{258.3}_{\pm5.2}$ |
| Timepilot | $14.9_{\pm14.3}$ | $19.3_{\pm31.0}$ | $18.3_{\pm38.1}$ | $6.61_{\pm7.5}$ | $124_{\pm26.1}$ | $\mathbf{341}_{\pm105}$ | $253_{\pm11.0}\bullet$ |
| Tutankham | $0.03_{\pm2.8}$ | $58.2_{\pm48.6}$ | $2.89_{\pm4.0}$ | $24.4_{\pm-0.4}$ | $91.6_{\pm29.3}$ | $130_{\pm10.7}\bullet$ | $\mathbf{134}_{\pm29.3}$ |
| Videopinball | $440_{\pm123}$ | $55.8_{\pm61.9}$ | $-4.03_{\pm32.5}$ | $626_{\pm241}$ | $299_{\pm168}$ | $\mathbf{1616}_{\pm1026}$ | $906_{\pm539}\bullet$ |
| **# Wins** | 0/15 | 0/15 | 0/15 | 0/15 | 0/15 | 6/15 | 9/15 |
| **# Super-Human** | 3/15 | 1/15 | 1/15 | 2/15 | 6/15 | **11/15** | **11/15** |

Table 3: Neural plasticity leads to vast performance improvements. Normalised mean scores and standard deviations (in percentage, *cf.* Appendix A.8 for the equation) of rigid baselines (*i.e.* DQN and DDQN with Leaky ReLU, DQN with SiLU and SiLU + dSiLU), as well as DQN with plasticity: using PELU, rational (full) and joint-rational (regularised), are reported over five experimental random seeded repetitions (larger mean values are better). The best results are highlighted in **bold** and runner-ups denoted with • markers. The last rows summarise the number of times best mean scores were obtained by each agent and the number of super-human performances.

**Final mean and maximum obtained scores of Rainbow agents:**

| Evaluation | **Final Mean Scores** | | | Max. Obtained Scores | | |
|---|---|---|---|---|---|---|
| Plasticity | rigid | full | regularised | rigid | full | regularised |
| Breakout | 52 | 279 | **303** | 383 | **569** | **569** |
| Enduro | 844 | **1473** | 1470 | 1388 | **1973** | 1964 |
| Kangaroo | 40 | **2157** | 2139 | **6300** | 6000 | 4800 |
| Q*bert | 149 | **11931** | 11551 | 16125 | **23550** | 23550 |
| Seaquest | 82 | 247 | **282** | 920 | **1280** | 1280 |
| Space Inv. | 595 | **1263** | 1157 | 2070 | **3395** | 2875 |
| Time Pilot | 3926 | 5386 | **6411** | 12700 | **15900** | 15900 |

Table 4: Final mean and maximum obtained scores obtained by rigid Rainbow agents (*i.e.* using Leaky ReLU), as well as Rainbow with full (*i.e.* using rational activation functions) and regularised (*i.e.* using joint-rational ones) plasticity (only 1 run because of computational cost, larger values are better).

**Final mean scores of all agents:**

| Algorithm | Random | DQN | | | DDQN | DQN with Plasticity | | |
|---|---|---|---|---|---|---|---|---|
| Network type | - | LReLU | SiLU | d+SiLU | LReLU | PELU | full | regularised |
| Asterix | $67.9_{\pm2.2}$ | $206_{\pm90}$ | $107_{\pm45}$ | $228_{\pm108}$ | $3723_{\pm1324}$ | $1998_{\pm275}$ | $\mathbf{18109_{\pm1755}}$ | $12621_{\pm2436}$ |
| Battlezone | $788_{\pm38}$ | $4464_{\pm2291}$ | $7612_{\pm4877}$ | $4429_{\pm2183}$ | $22775_{\pm11265}$ | $15807_{\pm6320}$ | $23403_{\pm701}$ | $\mathbf{25749_{\pm2837}}$ |
| Breakout | $0.14_{\pm01}$ | $155_{\pm46}$ | $26.2_{\pm16}$ | $3.4_{\pm3.89}$ | $79.4_{\pm33.8}$ | $219_{\pm22}$ | $315_{\pm36}$ | $\mathbf{336_{\pm10}}$ |
| Enduro | $0_{\pm0}$ | $121_{\pm158}$ | $274_{\pm131}$ | $2.77_{\pm3.41}$ | $353_{\pm134}$ | $181_{\pm315}$ | $\mathbf{1043_{\pm111}}$ | $957_{\pm109}$ |
| Jamesbond | $6.39_{\pm0.41}$ | $37.6_{\pm23.6}$ | $28.4_{\pm13.8}$ | $25.5_{\pm16.2}$ | $45.2_{\pm40.7}$ | $275_{\pm187}$ | $1122_{\pm176}$ | $\mathbf{1137_{\pm216}}$ |
| Kangaroo | $14.2_{\pm0.9}$ | $335_{\pm342}$ | $3500_{\pm2607}$ | $393_{\pm504}$ | $484_{\pm395}$ | $1586_{\pm398}$ | $2940_{\pm1175}$ | $\mathbf{5266_{\pm2365}}$ |
| Pong | $-20.2_{\pm0}$ | $15.9_{\pm2}$ | $14.1_{\pm4.3}$ | $16.9_{\pm1.2}$ | $12.4_{\pm11}$ | $17.8_{\pm0.8}$ | $18_{\pm0.9}$ | $\mathbf{18.1_{\pm1}}$ |
| Q*bert | $40.6_{\pm2.8}$ | $6715_{\pm2058}$ | $1754_{\pm2048}$ | $371_{\pm28}$ | $8954_{\pm2616}$ | $12143_{\pm795}$ | $\mathbf{14436_{\pm336}}$ | $14080_{\pm593}$ |
| Seaquest | $20.1_{\pm0.4}$ | $250_{\pm162}$ | $1504_{\pm1677}$ | $94.6_{\pm87.2}$ | $898_{\pm353}$ | $3740_{\pm991}$ | $6603_{\pm200}$ | $\mathbf{7461_{\pm1321}}$ |
| Skiing | $-16104_{\pm92}$ | $-27365_{\pm4794}$ | $-29890_{\pm4}$ | $-26725_{\pm5485}$ | $-26892_{\pm5881}$ | $-29912_{\pm10}$ | $\mathbf{-23487_{\pm7624}}$ | $-23582_{\pm7058}$ |
| Space Inv. | $51.6_{\pm1.1}$ | $531_{\pm62}$ | $520_{\pm169}$ | $509_{\pm176}$ | $490_{\pm15}$ | $759_{\pm48}$ | $650_{\pm45}$ | $\mathbf{1395_{\pm251}}$ |
| Tennis | $-23.9_{\pm0.0}$ | $-22.4_{\pm3.0}$ | $-19.4_{\pm9.2}$ | $-10.4_{\pm11.1}$ | $-18.4_{\pm8.9}$ | $-5.6_{\pm9.2}$ | $20.5_{\pm0.5}$ | $\mathbf{20.6_{\pm0.9}}$ |
| TimePilot | $688_{\pm30}$ | $1428_{\pm739}$ | $1644_{\pm1566}$ | $1594_{\pm1918}$ | $1016_{\pm401}$ | $6818_{\pm1323}$ | $\mathbf{17632_{\pm5242}}$ | $13261_{\pm576}$ |
| Tutankham | $3.51_{\pm0.54}$ | $3.55_{\pm4.3}$ | $81.9_{\pm66}$ | $7.41_{\pm5.96}$ | $36.4_{\pm0}$ | $127_{\pm40}$ | $179_{\pm15}$ | $\mathbf{184_{\pm40}}$ |
| VideoPinb. | $6795_{\pm461}$ | $45683_{\pm11383}$ | $11730_{\pm5941}$ | $6439_{\pm3336}$ | $62151_{\pm21791}$ | $42051_{\pm15356}$ | $\mathbf{149712_{\pm91219}}$ | $86942_{\pm48143}$ |

Table 5: Final mean raw scores (with std. dev.) of rigid baselines (*i.e.* DQN and DDQN with Leaky ReLU, DQN with SiLU and SiLU + dSiLU), as well as DQN with full plasticity (*i.e.* using rational activation functions) and regularised plasticity (*i.e.* using joint-rational ones) on Atari 2600 games, averaged over 5 seeded reruns (larger mean values are better).

**Maximum obtained scores:**

| Algorithm | Random | DQN | | | DDQN | DQN with Plasticity | | |
|---|---|---|---|---|---|---|---|---|
| Network type | - | LReLU | SiLU | d+SiLU | LReLU | PELU | full | regularised |
| Asterix | 71 | 9250 | 3400 | 3800 | 20150 | 9300 | 84950 | 49700 |
| Battlezone | 843 | 88000 | 81000 | 70000 | 97000 | 68000 | 78000 | 94000 |
| Breakout | 0 | 427 | 370 | 344 | 411 | 430 | 864 | 864 |
| Enduro | 0 | 1243 | 928 | 1041 | 1067 | 1699 | 1946 | 1927 |
| Jamesbond | 6 | 5600 | 5750 | 700 | 7500 | 6150 | 9250 | 13300 |
| Kangaroo | 15 | 14800 | 15600 | 10200 | 13000 | 12400 | 16200 | 16800 |
| Pong | -20 | 21 | 21 | 21 | 21 | 21 | 21 | 21 |
| Q*bert | 45 | 19425 | 11700 | 5625 | 19200 | 18900 | 24325 | 25075 |
| Seaquest | 20 | 7440 | 8300 | 740 | 15830 | 14860 | 9100 | 26990 |
| Skiing | -15997 | -5987 | -6505 | -6267 | -5359 | -5495 | -5368 | -5612 |
| Space Inv. | 53 | 2435 | 2205 | 2460 | 2290 | 2030 | 2490 | 3790 |
| Tennis | -23 | 8 | 1 | -1 | 4 | -1 | 24 | 36 |
| Time Pilot | 730 | 11900 | 15500 | 12500 | 12200 | 16300 | 72000 | 28000 |
| Tutankham | 4 | 249 | 267 | 267 | 274 | 397 | 334 | 309 |
| VideoPinb. | 7599 | 998535 | 950250 | 338512 | 991669 | 322655 | 997952 | 998324 |

Table 6: Maximum obtained scores (with std. dev.) of rigid baselines (*i.e.* DQN and DDQN with Leaky ReLU, DQN with SiLU and SiLU + dSiLU), as well as DQN with full plasticity (*i.e.* using rational activation functions) and regularised plasticity (*i.e.* using joint-rational ones) on Atari 2600 games, averaged over 5 seeded reruns (larger values are better).

**Human scores used for normalisation:**
Asterix: 7536, Battlezone: 33030, Breakout: 27.9, Enduro: 740.2, Jamesbond: 368.5, Kangaroo: 2739, Pong: 15.5, Q*bert: 12085, Seaquest: 40425.8, Skiing: $-3686.6$, Space Invaders: 1464.9, Tennis: $-6.7$, Time Pilot: 5650, Tutankham: 138.3, Video Pinball: 15641.1

## A.4 EVOLUTION OF THE SCORES ON EVERY GAME

The main part present some graphs that compares performance evolutions of the Rainbow and DQN agents with plasticity, as well as Rigid DQN, DDQN and Rainbow agents. We here provide the evolution of the scores of every tested DQN and the DDQN agents on the complete game set. DQN agents with higher plasticity are always the best-performing ones. Experiments on several games (e.g. Jamesbond, Seaquest) show that using DDQN does not prevent the performance drop but only delays it.

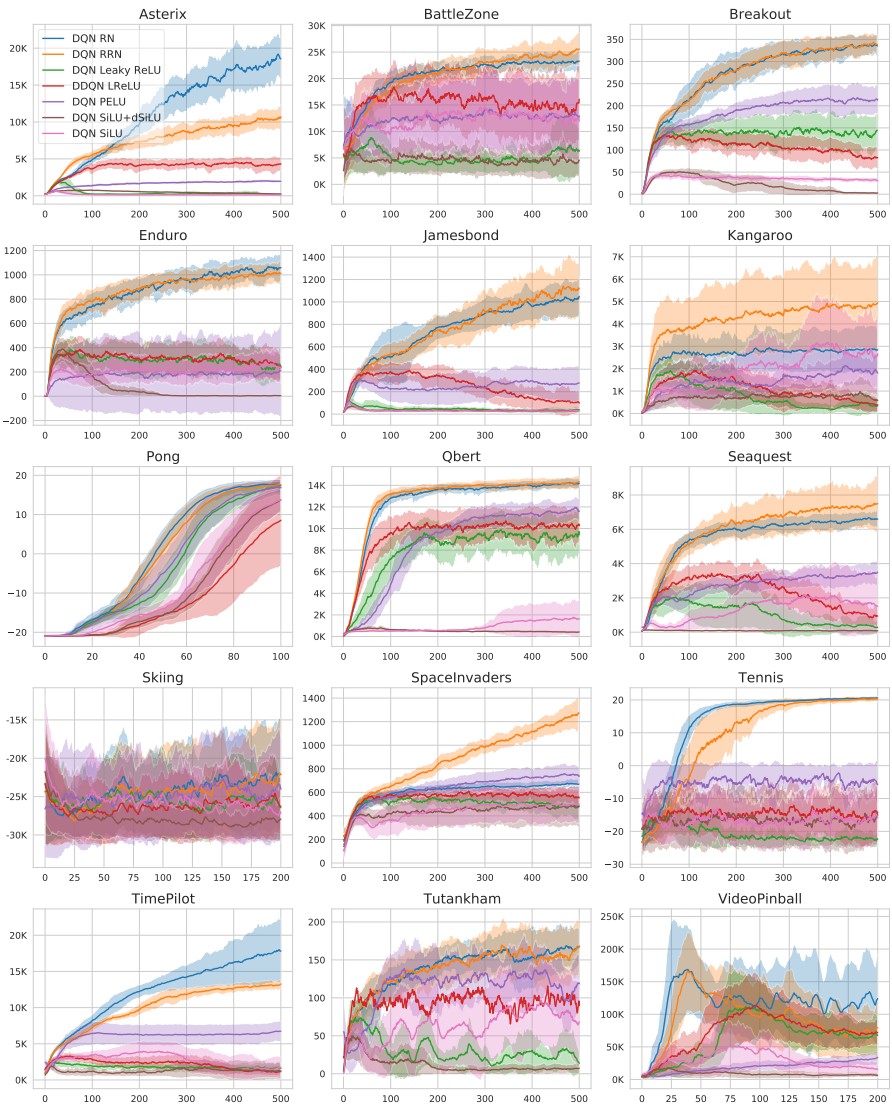

Figure 7: Smoothed (*cf.* Eq. 3) evolutions of the scores on every tested game for DQN agents with full (*i.e.* using rational activation functions) and regularised (*i.e.* using joint-rational ones) plasticity, and original DQN agents using Leaky ReLU, SiLU and SiLU+dSiLU, as well as for DDQN agents with Leaky ReLU.

## A.5 ENVIRONMENTS TYPES: STATIONARY, DYNAMICS AND PROGRESSIVE

The used environments have been separated in 3 categories, describing their potential changes through agents learning. This categorisation is here illustrated with frames of the tested games. As one can see: Breakout, Kangaroo, Pong, Skiing, Space Invaders, Tennis, Tutankham and VideoPinball can be categorised as **stationary environment**, as changes are minimal for the agents in these games. Asterix, BattleZone, Q*bert and Enduro present environment changes, that are early reached by the playing agents, and are thus **dynamic environments**. Finally, Jamesbond, Seaquest and Time Pilot correspond to **progressive environments**, as the agents needs to master early changes to access new parts of these environments.

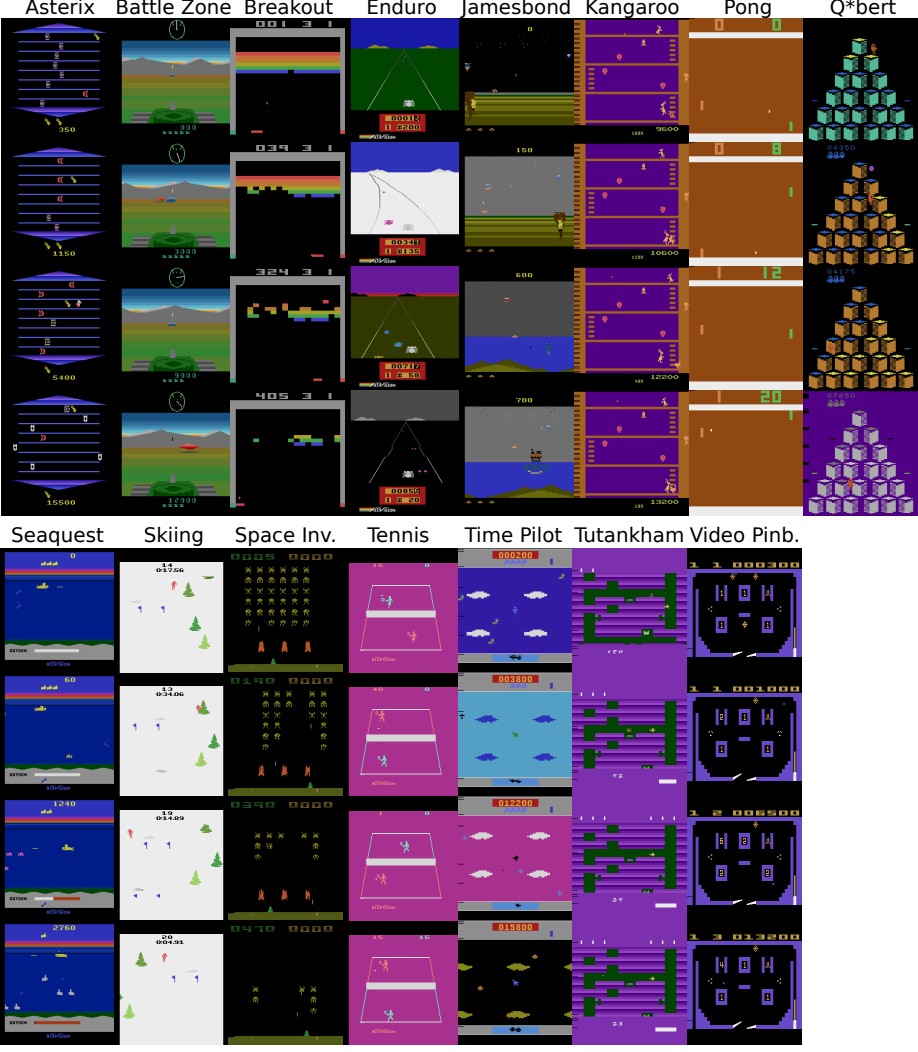

Figure 8: Images extracted from DQN agents with full plasticity playing the set of 15 Atari 2600 games used in this paper. Stationary environments (e.g. Pong, Video Pinball) do not evolve during training, dynamic ones provide different input/output distributions that are early accessible in the game (e.g Q*bert, Enduro) and progressive ones (e.g. Jamesbond, Time Pilot) require the agent to improve for the it to evolve.

## A.6 LEARNED RATIONAL ACTIVATION FUNCTIONS

We have explained in the main text how rational functions of agents used on different games can exhibit different complexities. This section provides the learned parametric rational functions learned by DQN agents with full plasticity (left) and by those with regularised plasticity (right) after convergence for every different tested game of the gym Atari 2600 environment. Kernel Density Estimations (with Gaussian kernels) of input distributions indicates where the functions are most activated. Rational functions from agents trained on simpler games (*e.g.* Enduro, Pong, Q*bert) have simpler profiles (*i.e.* fewer distinct extremas).

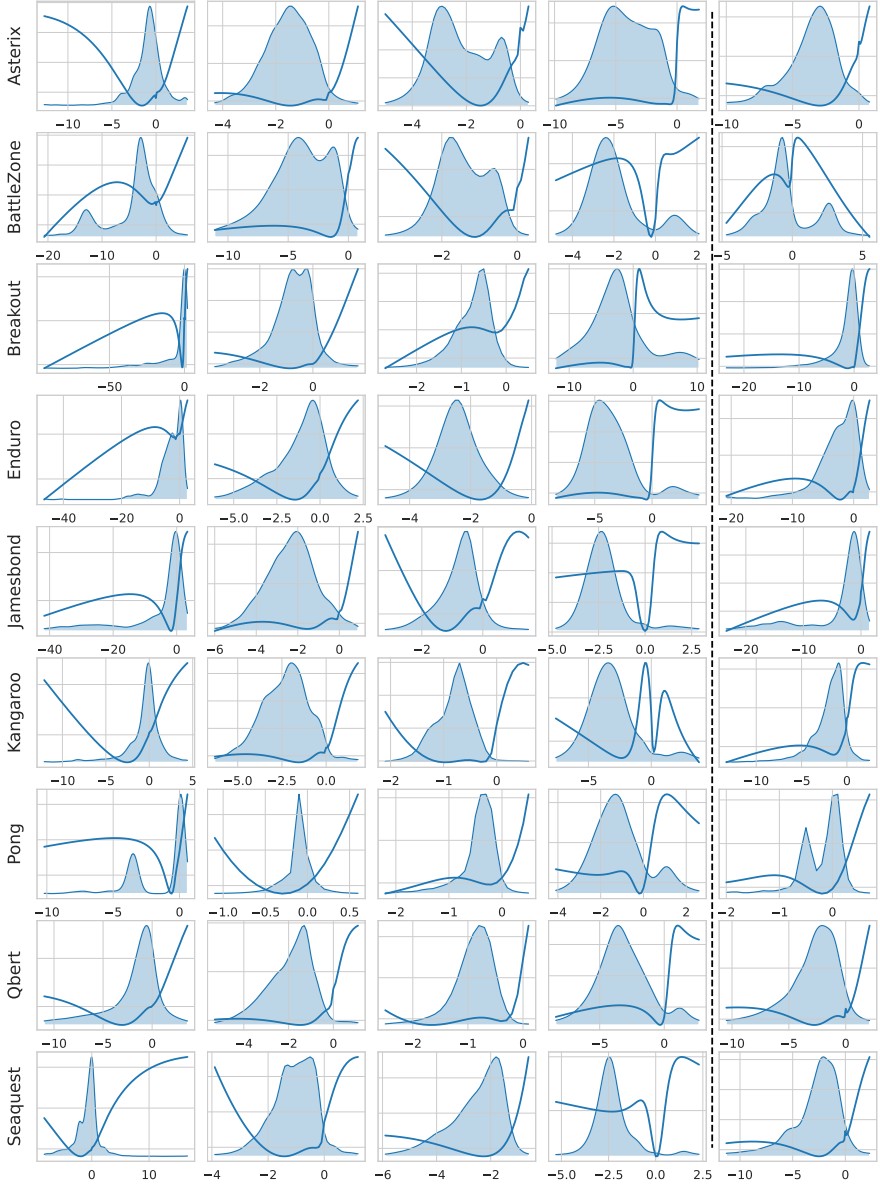

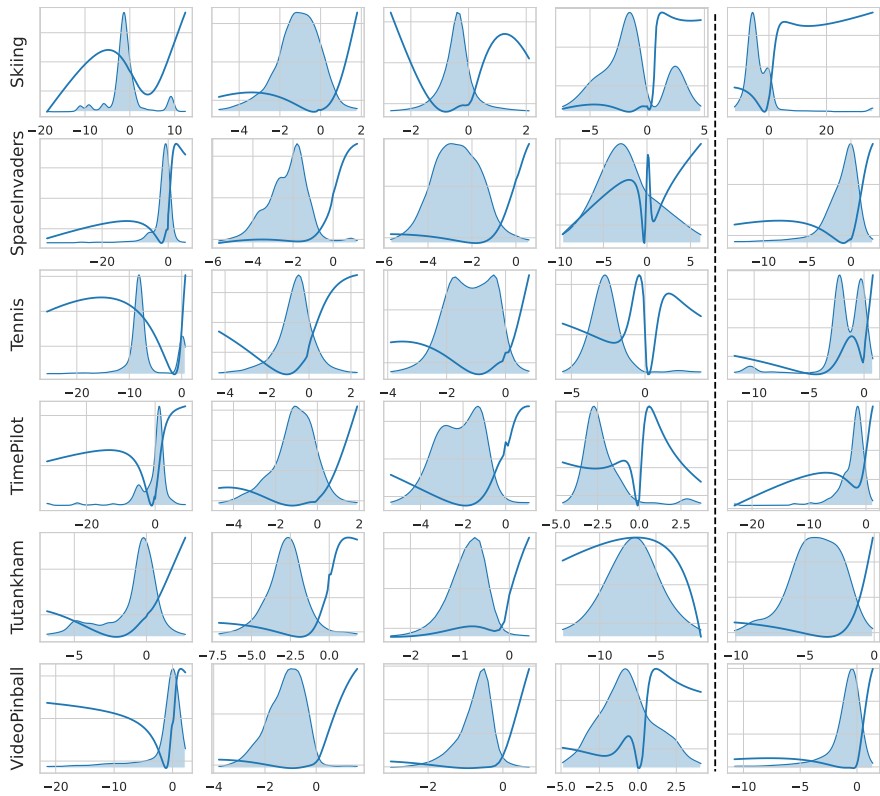

Figure 9: Profiles (dark blue) and input distributions (light blue) of rational functions (left) and joint-rational ones (right) of DQN agents on the different tested games. (Joint-)rational functions from agents of simpler games have simpler profiles (*i.e.* fewer distinct extrema).

## A.7 EVOLUTION OF RATIONALS ON THE PERM-MNIST CONTINUAL LEARNING EXPERIMENT

Figure 10 depicts the evolutions of rational functions through the permuted MNIST experiment. One can see that while the function of the first layer remains stable through the successive datasets, the second one flatten at its most activated region (around 0), while the third one increase its slope in this region, leading to higher gradients. This suggests that rational functions can help adapting the gradient scales at each layer. Further investigating this is an interesting line of future work.

Figure 10: Evolution of the rational activation functions on the permuted MNIST experiment (*cf.* 4). The 3 rational activation functions used for training (and retraining) are adapting to fit the data (depicted in semi transparent).

## A.8 TECHNICAL DETAILS TO REPRODUCE THE EXPERIMENTS

We here provide details on our experiments for reproducibility. **We used the seed 0, 1, 2, 3, 4 for every multi-seed experiment.**

### SUPERVISED LEARNING EXPERIMENTS

For the lesioning experiment, we used an available[5] pretrained Residual Network. We then remove the corresponding block (and potentially replace it with an identity initialised rational activation function) (surgered). We finetune the new models, allowing for optimisation of the previous and next layers (and potentially the rational function) for 15 epochs with SGD (learning rate of 0.001).

---

[5]https://download.pytorch.org/models/resnet101-5d3b4d8f.pth

For the classification experiments, we run on CIFAR10 and CIFAR100 (Krizhevsky et al., MIT License), we let every network learn for 60 epochs. We use the code provided by Molina et al. (2020), with only one classification layer in these smaller VGG versions (VGG4, VGG6 and VGG8, against 3 for VGG16 and larger). We use SGD as the optimisation algorithm, with a learning rate of $0.02$ and $128$ as batch size. The VGG networks contain successive VGG blocks that all consist of $n$ convolutional layers, $i$ input channels and $o$ output channels, stride 3 and padding 1, followed by an activation function, and 1 Max Pooling layer. For each used architecture, the $(n, i, o)$ parameters of the successive blocks are:

- VGG4: $(1, 3, 64) \rightarrow (1, 64, 128) \rightarrow (2, 128, 256)$
- VGG6: $(1, 3, 64) \rightarrow (1, 64, 128) \rightarrow (2, 128, 256) \rightarrow (2, 256, 512)$
- VGG8: $(1, 3, 64) \rightarrow (1, 64, 128) \rightarrow (2, 128, 256) \rightarrow (2, 256, 512) \rightarrow (2, 512, 512)$

The output of these blocks is then passed on to a classifier (linear layer). Only activation functions differ between the Leaky ReLU and the Rational versions.

REINFORCEMENT LEARNING EXPERIMENTS

To ease the reproducibility of our the reinforcement learning experiments, we used the *Mushroom RL* library (D'Eramo et al., 2020) on the Arcade Learning Environment (GNU General Public License). We used states consisting of 4 consecutive grey-scaled images, downsampled to $84 \times 84$. Computing the gradients for rational functions takes longer than e.g. ReLU. However, we used a CUDA optimized implementation of the rational activation functions that we open source along with this paper. In practice, we did not notice any significant training time difference.

**Network Architecture.** The input to the network is thus a 84x84x4 tensor containing a rescaled, and gray-scaled, version of the last four frames. The first convolution layer convolves the input with 32 filters of size 8 (stride 4), the second layer has 64 layers of size 4 (stride 2), the final convolution layer has 64 filters of size 3 (stride 1). This is followed by a fully-connected hidden layer of 512 units. All these layers are separated by the corresponding activation functions (either Leaky ReLU, SiLU, SiLU for convolution layers and dSiLU for linear ones, PELU, rational functions (at each layer) and joint-rational ones (shared accross layers) of order $m = 5$ and $n = 4$, initialised to approximate Leaky ReLU). We used the default PeLU initial hyperparameters (a=1, b=1, c=1) and let the weights optimizer tune them through training, as for rational functions. For CRELU, we took the implementation from ML Compiled [6], and halves the number of filters in the following convolutional layers to keep the same network structure intact, as done by Shang et al. (2016).

**Hyper-parameters.** We evaluate the agents every 250K steps, for 125K steps. The target network is updated every 10K steps, with a replay buffer memory of initial size 50K, and maximum size 500K, except for Pong, for which all these values are divided by 10. The discount factor $\gamma$ is set to 0.99 and the learning rate is 0.00025. We do not select the best policy among seeds between epochs. We use the simple $\epsilon$-greedy exploration policy, with the $\epsilon$ decreasing linearly from 1 to 0.1 over 1M steps, and an $\epsilon$ of 0.05 is used for testing.

The only difference from the evaluation of Mnih et al. (2015) and of van Hasselt et al. (2016) evaluation is the use of the Adam optimiser instead of RMSProp, for every evaluated agent.

**Normalisation techniques.** To compute human normalised scores, we used the following equation:

$$\text{score}_{\text{normalised}} = 100 \times \frac{\text{score}_{\text{agent}} - \text{score}_{\text{random}}}{\text{score}_{\text{human}} - \text{score}_{\text{random}}}, \tag{2}$$

For readability, the curves plotted in the Fig. 4 and Fig. 8 are smoothed following:

$$score_t = \alpha \times score_{t-1} + (1 - \alpha) \times scores_t, \tag{3}$$

with $\alpha = 0.9$.

**Overestimation computation**. We used the following formulae to compute relative overestimation.

$$\text{overestimation} = \frac{\text{Q-value} - R}{R} \tag{4}$$

---

[6]https://ml-compiled.readthedocs.io/en/latest/activations.html

RL NETWORK ARCHITECTURE

The DQN, DDQN and Rainbow agents networks architecture, rational plasticity (with rational activations functions at each layer) and of the regularized ones (with one joint-rational activation function shared across layers). For the other activation functions, the "Rat." blocks are replaced with Leaky ReLU, CReLU, SiLU, or PELU. For the d+SiLU networks, SiLU is used on the convolutional layers (*i.e.* first two), and dSiLU in the fully connected ones (*i.e.* last two).

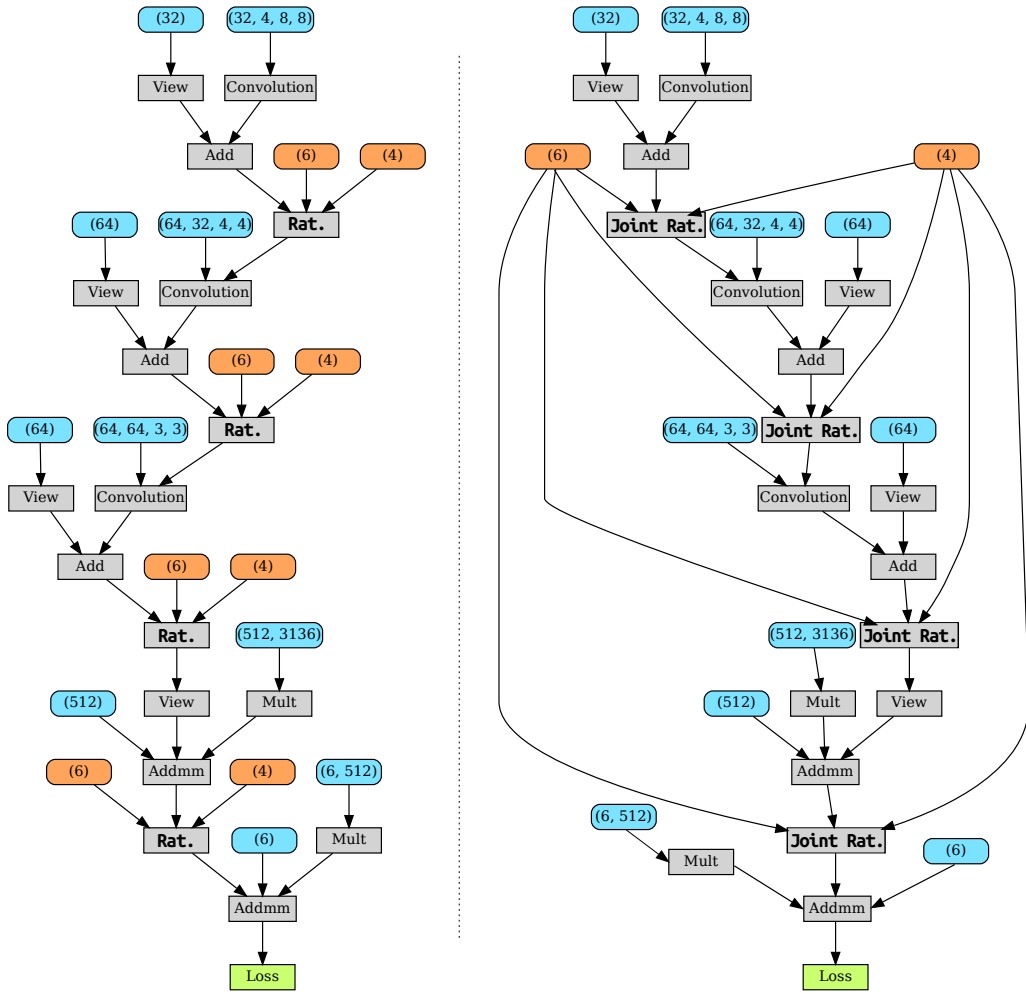

Figure 11: left: The DQN agents' neural network equipped with Rational Activation Functions (Rat.). Any other network with classical activation functions (as Leaky Relu or SiLU) would be similar, with the corresponding activation function instead of the rational one. right: The agents' network using the regularized joint-rational version of the network. The same activation is used across the layers. The parameters of the rational activation (in orange) function are shared. In both graphs, operations are placed in the grey boxes and parameters in the blue ones, (or orange for the rationals' ones).

