| $1.85_{\pm 1.2}$ | $0.52_{\pm 0.6}$ | $2.14_{\pm 1.4}$ | $48.9_{\pm 17.7}$ | $25.8_{\pm 3.7}$ | $\mathbf{242}_{\pm 23.5}$ | $168_{\pm 32.6}\bullet$ |
| Battlezone | $11.4_{\pm 7.0}$ | $21.2_{\pm 15.0}$ | $11.3_{\pm 6.7}$ | $68.2_{\pm 34.8}$ | $46.6_{\pm 19.5}$ | $70.1_{\pm 2.1}\bullet$ | $\mathbf{77.4}_{\pm 8.7}$ |
| Breakout | $558_{\pm 166}$ | $93.9_{\pm 57.6}$ | $11.7_{\pm 14.0}$ | $286_{\pm 122}$ | $788_{\pm 79.2}$ | $1134_{\pm 130}\bullet$ | $\mathbf{1210}_{\pm 36.0}$ |
| Enduro | $16.3_{\pm 21.3}$ | $37.0_{\pm 17.7}$ | $0.37_{\pm 0.5}$ | $47.7_{\pm 18.1}$ | $24.5_{\pm 42.6}$ | $\mathbf{141}_{\pm 15.0}$ | $129_{\pm 14.7}\bullet$ |
| Jamesbond | $8.62_{\pm 6.4}$ | $6.08_{\pm 3.7}$ | $5.28_{\pm 4.4}$ | $10.7_{\pm 11.1}$ | $74.2_{\pm 51.5}$ | $308_{\pm 48.5}\bullet$ | $\mathbf{312}_{\pm 59.5}$ |
| Kangaroo | $11.8_{\pm 12.5}$ | $128_{\pm 95.6}\bullet$ | $13.9_{\pm 18.5}$ | $17.2_{\pm 14.5}$ | $57.7_{\pm 14.6}$ | $107_{\pm 43.1}$ | $\mathbf{193}_{\pm 86.8}$ |
| Pong | $101_{\pm 5.5}$ | $96.1_{\pm 12.0}$ | $104_{\pm 3.3}$ | $91.3_{\pm 30.8}$ | $106.4_{\pm 2.2}$ | $107.0_{\pm 2.4}\bullet$ | $\mathbf{107.3}_{\pm 2.7}$ |
| Qbert | $55.4_{\pm 17.1}$ | $14.2_{\pm 17.0}$ | $2.74_{\pm 0.2}$ | $74.0_{\pm 21.7}$ | $101_{\pm 6.6}$ | $\mathbf{120}_{\pm 2.8}$ | $117_{\pm 4.9}\bullet$ |
| Seaquest | $0.57_{\pm 0.4}$ | $3.67_{\pm 4.1}$ | $0.18_{\pm 0.2}$ | $2.17_{\pm 0.9}$ | $9.21_{\pm 2.5}$ | $16.3_{\pm 0.5}\bullet$ | $\mathbf{18.4}_{\pm 3.3}$ |
| Skiing | $-90.7_{\pm 37.9}$ | $-111_{\pm -0.7}$ | $-85.5_{\pm 43.4}$ | $-86.9_{\pm 46.6}$ | $-111_{\pm -.7}$ | $\mathbf{-59.5}_{\pm 60.7}$ | $-60.2_{\pm 56.1}\bullet$ |
| Space Inv. | $33.9_{\pm 4.3}$ | $33.1_{\pm 11.9}$ | $32.4_{\pm 12.4}$ | $31.0_{\pm 1.0}$ | $50.1_{\pm 3.3}\bullet$ | $42.3_{\pm 3.1}$ | $\mathbf{95.1}_{\pm 17.7}$ |
| Tennis | $8.94_{\pm 17.3}$ | $26.3_{\pm 53.3}$ | $78.5_{\pm 64.3}$ | $32.1_{\pm 51.6}$ | $106_{\pm 53.3}$ | $257.8_{\pm 2.8}\bullet$ | $\mathbf{258.3}_{\pm 5.2}$ |
| Timepilot | $14.9_{\pm 14.3}$ | $19.3_{\pm 31.0}$ | $18.3_{\pm 38.1}$ | $6.61_{\pm 7.5}$ | $124_{\pm 26.1}$ | $\mathbf{341}_{\pm 105}$ | $253_{\pm 11.0}\bullet$ |
| Tutankham | $0.03_{\pm 2.8}$ | $58.2_{\pm 48.6}$ | $2.89_{\pm 4.0}$ | $24.4_{\pm -0.4}$ | $91.6_{\pm 29.3}$ | $130_{\pm 10.7}\bullet$ | $\mathbf{134}_{\pm 29.3}$ |
| Videopinball | $440_{\pm 123}$ | $55.8_{\pm 61.9}$ | $-4.03_{\pm 32.5}$ | $626_{\pm 241}$ | $299_{\pm 168}$ | $\mathbf{1616}_{\pm 1026}$ | $906_{\pm 539}\