# OpenReview forum: "Adaptive Rational Activations to Boost Deep Reinforcement Learning"
_ICLR.cc/2024/Conference — ICLR 2024 spotlight_

### Official Review · Reviewer_qikv · 2023-10-21

**Soundness:** 2 fair
**Presentation:** 1 poor
**Contribution:** 3 good
**Rating:** 6
**Confidence:** 3

**Summary:**

The authors study the use of rational activation functions for RL.  They show that these functions embed residual connections.  They study them empirically.

**Strengths:**

An interesting, well-motivated idea, with a large number of various, potentially significant contributions.

**Weaknesses:**

**Small edits and minor recommendations:**

- "emphasise" spelling error in 2.1
- In the “Network Architecture” section of the appendix, there is a parentheses mismatch.
- Does “resp.” mean respectively?  This stumped me for a couple minutes, I suggest being more clear and not using abbreviations like this, or at least defining them before use.
- 2.3: “However, if we circle back to this figure” Which figure? Be more clear; don’t make your reader work unnecessarily hard to understand.

**Clarity Weaknesses:**
- 2.1: the authors say very little about the architecture.  Is it conv nets?  Is the number of units between the baselines and their approach fixed, and they replace each unit’s activation function with a rational plasticity unit (resulting in many more parameters for their approach)?  Or do they replace whole layers with a single rational plasticity unit (the parenthetical in 2.1 seems to hint at this, but it’s unclear)?  Or do they keep all units from each layer intact, and add a singe additional rational plasticity unit (if I recall correctly, this is what the original rational activation function paper did)?  These things are critically important pieces of information about this paper’s results (this is essentially an architecture paper, but the reader is left guessing about the most basic aspects of the architectures used), and this critical information should not be relegated to an appendix.  Worse, the last last question (how exactly rational plasticity units replace traditional units in an architecture) is unclear to me, even after reading the Network Architecture section of the appendix.  Update on this last: more details are given, almost parenthetically, on page 6 and again on page 8 (in the Q4 discussion).  However, the long delay hinders clarity, and even with these details, the exact details of the approach/architecture remain unclear.  I suggest clarifying all of these questions in 2.1; a figure could be very helpful.

- “To this end, we also show agents equipped with rational activations that are tempered in Fig. 2. They have been extracted from agents with rational plasticity, that adapted to their specific environment. The plasticity of the rational activation functions is then tempered (“stopped”) in repeated application to emphasise the necessity to continuously adapt during training.” I am not sure exactly what this means; it is too vague and can think of several different interpretations with very different meanings.  In the second sentence, it sounds like they freeze these rational activations after a while, but let the other network weights continue to be optimized.  But then the third sentence seems to contradict this (what does “repeated application” mean, and why would freezing some parameters “emphasize the necessity to continuously adapt”?)

- 2.2 Theorem concern: The precise meaning of this theorem is unclear.  As stated, it can easily be taken to mean that all rational activation functions implicitly contain a residual connection.  But if I’m understanding the proof correctly, it seems to indicate that it depends on the weights; that is, that a residual connection can be learned if necessary.  This is confusing and potentially misleading.  I suggest stating the Theorem more precisely.

- 2.3: “regularized joint-rationals, where the key idea is to constraint the input to propagate through different layers but always be activated by the same learnable rational activation function”.  This is incredibly vague, and does not tell the reader what this approach means (and this sentence seems to constitute the entire description of joint rationals).

**Other weaknesses:**
- Is plasticity actually causing the better results?  Q3 is a very nice analysis, and provides evidence that the proposed approach reduces overestimation.  However, it is not clearly demonstrated that: 1) the plasticity in particular is what causes this reduction in overestimation OR 2) that the reduction in overestimation is what causes the improved performance in Q1 and Q2. BOTH of these things would need to be shown to support the authors’ claims.  In fact, Figure 2 shows evidence against the former: rational activation functions in general seem to simply perform better even for stationary environments, which is nice, but seems to contradict the core thesis of the paper (rational activation functions cause better plasticity which in turn causes better performance).
- The above is even more concerning in light of the lack of details about architectures used.  Perhaps the better performance is simply due to their approach using networks with more parameters and thus more capacity?  Unfortunately, it is not even clear how the different approaches’ architectures vary, so it is difficult for a reader to even begin to evaluate this concern.


Note: Despite the many significant concerns I have about this paper, I think it already has many nice results and strengths as well, in addition to being a well-motivated work.  If a rating of 4 were an option, I would give it a 4 instead of a 3.  I encourage the authors to work to address the clarity issues and the "Is plasticity actually causing the better results?" concern, and resubmit; I think this is a potentially very strong contribution.

**Questions:**

See clarity concerns above.

---

> ### Author Response · Authors · 2023-11-22
> **We appropriate your help on the clarity of the manuscript, please consider our updated version.**
>
> Dear reviewer, thank you very much for the time you have spent, helping us to improve our manuscript. An updated version has been uploaded, for which all your feedback have been considered and integrated. These modifications are done in blue in our updated manuscript. Please find the detailed points hereafter.
>
> **Small edits.** We appreciate these pointers, we corrected all of these typos in the updated manuscript.
>
> **Clarity issues:**
>
> **Description of the architecture**. We use the same algorithm and network architecture as used in the previous DQN, DDQN and Rainbow papers. The code is open source and the implementation details are in Appendix A9. As these details concern the RL experiments only, we added another reference to it in the first paragraph of section 3.
>
> For the rational activation functions, they are used layer-wise, as in the original paper [1], or are even shared across layers for the regularized version, adding only 10 or 40 parameters to the overall network. We agree that these details should be provided upfront. We now give more precise descriptions in 3rd paragraph of the introduction, in paragraph 2.3 and in Appendix A9 as it is indeed crucial to understand our contribution. Thank you for this comment. We also provide graphs of the network architectures for rational plasticity and regularized rational plasticity in appendix (cf Figure 11).
>
> **Tempered plasticity.** We modified this paragraph to make improve its clarity, with: \
> “To this end, we show agents equipped with rational activations that are tempered in Fig.2.
>
> Such agents are equipped with the final, optimised rational functions of trained rational-equipped agents. They correspond to frozen functions from agents that already adapted to their specific environment. The plasticity of the rationals is thus tempered (i.e. stopped) in a repeated application (i.e. another training session) to emphasise the necessity to continuously adapt the activation functions, together with the layers' weights during training.”.
>
> Freezing hyperparameters show that providing agents with pretrained fixed activation functions, that are tailored to each environment, helps, but not as much as providing plastic activation functions (i.e. functions that continuously adapt during training).
>
> **Theorem in 2.2.** We have modified the formulation of the theorem 2.2 as well as reformulated its descriptions and explanations. It now clearly states that rational functions include a residual connection _if and only if_ the numerators’ degree is higher than the denominator’s one.
>
> **Description of joint-rationals.** We changed our description to:
>
> “We propose the regularized joint-rationals, where we constrain the input to propagate through different layers but always be activated by the same learnable rational activation function. Rational functions thus share a mutual set of parameters across the network (instead of layers, cf. Fig. 11)).”, with the newly provided Fig. 11 also helps to understand the integration of joint-rationals.
>
> **Is plasticity the cause of the improvements?** Providing neural networks with activation functions able to continuously adapt, through time, to the fitted data, augment their plasticity (in comparison to rigid ones), by definition of plasticity. To even further prove our point, we now show that rational activation functions improve results of the permuted MNIST experiment (cf. Appendix A1). In this experiment, the network is trained on the MNIST dataset, then on variations of these image where different fixed permutation functions are applied. The experiment shows that networks equipped with rationals are both better at learning the current tasks, and retaining information from the previous one.
>
> Moreover, as explained in Q3), overestimation in deep Q learning has been attributed to a lack of plasticity of the function by the authors of DDQN [2], who instead of solving this plasticity problem, proposed a trick to mitigate the propagation of the overestimated states.
>
> The fact that rational as activation functions augment plasticity is now clear, both because of the definition of plasticity and our set of experiments. The fact it reduces overestimation is clearly demonstrated in Q3, and the fact that reducing overestimation improves the agent performances has already been proven many times. We hope that potential confusion are addressed.

---

> ### Author Response · Authors · 2023-11-22
> **Short follow up**
>
> **The performance improvement might be due to the extra parameters.** This is exactly what Q4 ("***How many more parameters would rigid networks need to measure up to rational ones?"***) is addressing. We have provided more details about the used architectures. As explained in the Q4 first paragraph, if extra parameters would be the only helping factors, then agents equipped with different PELU functions at each layer would outperform (or at least perform on par with) the one with equipped with joint-rationals. PELU networks embed 12 more parameters, against 10 for joint-rationals ones, network-wise.
> To further prove that parameters are not enough, we even conducted the experiments answering Q4, showing that rigid baselines need up to 3.5 times as many parameters as the architectures that use rational functions in order to obtain similar performances.
>
> We think that the improved clarity now avoids any interrogation concerning these questions.
>
> Thank you for helping us with these issues.

---

### Official Review · Reviewer_mxps · 2023-10-30

**Soundness:** 3 good
**Presentation:** 2 fair
**Contribution:** 3 good
**Rating:** 8
**Confidence:** 3

**Summary:**

Neural plasticity has been shown to be crucial to reinforcement learning (RL), especially to environments that change constantly and/or rapidly. This work proposes the use of adaptive activation functions to increase plasticity. Particualrly, it employs a form of rational activation functions, i.e., joint-rational activation, to have greater flexibility while avoiding overfitting. Experimentally, the work shows that such functions can make DQN competitive with DDQN and Rainbow without all the additional techniques.

**Strengths:**

1. The work tackles the important problem of plasticity in deep RL with a novel method based on adaptive activation functions.
2. Showing using rational functions can compensate for residual block is an intriguing and insightful finding

**Weaknesses:**

1. Missing related works in the study of plasticity in deep RL (e.g., [1])
2. There are works out there that tackles plasticity in deep RL. They should be added as baselines to compare with the proposed method (e.g., [2], [3])
3. Results would be a lot stronger if it can show that the results transfer across algorithms too (e.g., policy gradient methods)

[1] Abbas, Z., Zhao, R., Modayil, J., White, A., & Machado, M. C. (2023). Loss of plasticity in continual deep reinforcement learning. arXiv preprint arXiv:2303.07507.

[2] Nikishin, E., Oh, J., Ostrovski, G., Lyle, C., Pascanu, R., Dabney, W., & Barreto, A. (2023). Deep Reinforcement Learning with Plasticity Injection. arXiv preprint arXiv:2305.15555.

[3] Dohare, S., Sutton, R. S., & Mahmood, A. R. (2021). Continual backprop: Stochastic gradient descent with persistent randomness. arXiv preprint arXiv:2108.06325.

**Questions:**

1. I am not very familiar with the area of adaptive activation functions. But what in particular do rational activation functions offer that make it superior over some of the options out there? E.g., I came across [1]
2. Can the authors provide some intuition on why improved plasticity lead to reduced overestimation?

Jagtap, A. D., Kawaguchi, K., & Karniadakis, G. E. (2020). Adaptive activation functions accelerate convergence in deep and physics-informed neural networks. Journal of Computational Physics, 404, 109136.

---

> ### Author Response · Authors · 2023-11-22
> **Thank you, we modified our manuscript following your feedback**
>
> Dear Reviewer,
>
> we thank you for the valuable feedback. We really appreciate all the time you have already spent to help us improve our paper. The points raised have helped us to improve.
>
> We updated manuscript, and highlight the modifications done for clarity.
>
> Let us now shortly address the opportunities.
>
> **Missing literature on plasticity.** Sorry for this. We added an entire paragraph called **_Plasticity in deep RL_** in our related work section, discussing all the work you and other reviewers have provided us with. In our Fig. 1, we have now added results of agents that use CReLU and commented on it.
>
> **Show that results transfer across algorithms.** To show that our results are algorithm agnostic, we have our experiment on Rainbow. We understand that bringing rational activation functions on policy learning algorithms would be helpful as well, but it is unexpected that providing such agents with rational plasticity does not hold true for e.g. PPO, as . All the recommended paper on plasticity in RL only use one algorithm to prove their point [1, 2, 3, 4, 5, 6, 7], with [7] showing that augmented plasticity boosts PPO agents.
>
> **Questions:**
>
> **Why are rationals superior to other existing learnable functions?** As explained in the introduction and related work sections, rational functions, similar to polynomials are universal approximants, not weighted combinations of existing functions, that they can all approximate [8], making them even more flexible or adaptive than these functions. They were also already showed to improve performances over all these existing functions on classification tasks [8], and shown to be better approximants than polynomials [9]. We have modified our text to make these points clearer (cf Introduction and Related Work).
>
> **Why does rational plasticity help with overestimation?** The authors of the DDQN paper [10] explain and show that inaccurate plasticity leads to overestimation, then augmented via the _max_ operator of the DQN equation. They mitigate this augmentation with the double q learning technique. As shown through the paper, rationals provide adaptive plasticity, converging to complicated approximation functions for complicated games (c.f. BattleZone, Kangaroo, SpaceInvaders in Fig 9 compared to e.g. Pong and Qbert). The intuition is that the network is used to predict the value function. As depicted in Fig. 2 of [10], under or over-fitting the state/action space leads to higher values for interpolated points (i.e., less visited states).
>
> [1] Ghada Sokar, Rishabh Agarwal, Pablo Samuel Castro, and Utku Evci. The dormant neuron phenomenon in deep reinforcement learning. arXiv preprint arXiv:2302.12902, 2023.
>
> [2] Zaheer Abbas, Rosie Zhao, Joseph Modayil, Adam White, and Marlos C Machado. Loss of plasticity in continual deep reinforcement learning. arXiv preprint arXiv:2303.07507, 2023.
>
> [3] Clare Lyle, Zeyu Zheng, Evgenii Nikishin, Bernardo Avila Pires, Razvan Pascanu, and Will Dabney. Understanding plasticity in neural networks. arXiv preprint arXiv:2303.01486, 2023.
>
> [4] Evgenii Nikishin, Max Schwarzer, Pierluca D’Oro, Pierre-Luc Bacon, and Aaron Courville. The primacy bias in deep reinforcement learning. In International Conference on Machine Learning, 2022.
>
> [5] Nikishin, E., Oh, J., Ostrovski, G., Lyle, C., Pascanu, R., Dabney, W., & Barreto, A. (2023). Deep Reinforcement Learning with Plasticity Injection. arXiv preprint arXiv:2305.15555.
>
> [6] Abbas, Z., Zhao, R., Modayil, J., White, A., & Machado, M. C. (2023). Loss of plasticity in continual deep reinforcement learning. arXiv preprint arXiv:2303.07507.
>
> [7] Dohare, S., Sutton, R. S., & Mahmood, A. R. (2021). Continual backprop: Stochastic gradient descent with persistent randomness. arXiv preprint arXiv:2108.06325.
>
> [8] Molina, Alejandro et al. “Padé Activation Units: End-to-end Learning of Flexible Activation Functions in Deep Networks.” _ArXiv_ abs/1907.06732 (2019)
>
> [9] Telgarsky, Matus. “Neural Networks and Rational Functions.” _ArXiv_ abs/1706.03301 (2017)
>
> [10] Hasselt, H. V. et al. “Deep Reinforcement Learning with Double Q-Learning.” _AAAI Conference on Artificial Intelligence_ (2015).

---

> > ### Comment · Reviewer_mxps · 2023-11-23
> >
> > Thank you for your response. With the additional experiments, clarifications and updates to related work, my concerns have been addressed. In agreement with reviewer 7Gju, I think this is a good paper and recommend acceptance. I have raised my scores accordingly.

---

> > > ### Author Response · Authors · 2023-11-23
> > > **Thank you for your answer and revised score.**
> > >
> > > Dear reviewer, thank you very much for spending time again on our answer and updated manuscript.
> > > We are very pleased that your concerned have been addressed.
> > > Thank you very much for the time you have again devoted to our work.

---

### Official Review · Reviewer_7Gju · 2023-10-31

**Soundness:** 3 good
**Presentation:** 2 fair
**Contribution:** 3 good
**Rating:** 8
**Confidence:** 4

**Summary:**

The manuscript argues that plasticity is important for RL. The paper proposes rational activation functions for RL. It also proposes a regularized version of the activation function (joint rational). The results show that rational activation functions significantly outperform rigid activation functions like leaky-ReLU.

**Strengths:**

The idea of using rational activation functions to inject plasticity is very interesting, and the results are promising. DQN with rational activation functions significantly outperforms vanilla DQN. The use of rational activation in RL seems novel. I'm not aware of any work that uses these activations in RL. Most of the paper is clearly written and easy to understand.

**Weaknesses:**

Although the paper contains exciting ideas and promising results, there are many significant weaknesses in the paper that stop me from recommending a full acceptance.
- **Missing literature on plasticity.** The manuscript needs to include the entire literature on plasticity. There are already many papers that show that there is a loss of plasticity when deep learning systems face a non-stationary data stream, such as in RL (Dohare et al., 2021; Lyle et al., 2022; Nikishin et al., 2022; Abbas et al., 2023; Lyle et al., 2023; Sokar et al., 2023; Nikishin et al., 2023; Dohare et al., 2023). Missing this literature also means that the paper is missing important baselines in the paper methods like CReLUs, selective reinitialization, layer normalization, etc. I realize that doing a full comparison in Atari with all of these methods might not be computationally feasible, but they should be compared in a smaller problem.
- **Unsupported conclusions.** The manuscript claims that rational activation functions help because they inject plasticity. However, the paper does not provide any direct evidence of this claim. The paper shows that deep RL methods with rational activations outperform deep RL methods with rigid activations. Better performance does not mean better plasticity. However, this problem can be somewhat reduced by comparing rational and rigid activations in a continual supervised learning problem like random label MNIST (Lyle et al., 2023) or permuted MNIST (Dohare et al., 2023). Then, we will have direct evidence that rational activations inject plasticity.

Dohare, S., Sutton, R. S., & Mahmood, A. R. (2021). Continual backprop: Stochastic gradient descent with persistent randomness. arXiv preprint arXiv:2108.06325.

Nikishin, E., Schwarzer, M., D’Oro, P., Bacon, P.-L. & Courville, A. The primacy bias in deep reinforcement learning. In International Conference on Machine Learning, 16828–16847 (PMLR, 2022).

Lyle, C., Rowland, M. & Dabney, W. Understanding and preventing capacity loss in reinforcement learning. In
International Conference on Learning Representations (2022).

Abbas, Z., Zhao, R., Modayil, J., White, A., & Machado, M. C. (2023). Loss of plasticity in continual deep reinforcement learning. CoLLAs 2023.

Sokar, G., Agarwal, R., Castro, P. S. & Evci, U. The dormant neuron phenomenon in deep reinforcement learning. In Krause, A. et al. (eds.) Proceedings of the 40th International Conference on Machine Learning, vol. 202 of Proceedings of Machine Learning Research, 32145–32168 (PMLR, 2023).

Lyle, C. et al. Understanding plasticity in neural networks. In Krause, A. et al. (eds.) Proceedings of the 40th International Conference on Machine Learning, vol. 202 of Proceedings of Machine Learning Research, 23190–23211 (PMLR, 2023).

Nikishin, E., Oh, J., Ostrovski, G., Lyle, C., Pascanu, R., Dabney, W., & Barreto, A. (2023). Deep Reinforcement Learning with Plasticity Injection. arXiv preprint arXiv:2305.15555.

Dohare, S., Hernandez-Garcia, J. F., Rahman, P., Sutton, R. S., & Mahmood, A. R. (2023). Loss of Plasticity in Deep Continual Learning. arXiv preprint arXiv:2306.13812.

**Questions:**

Can you please clarify what exactly is the rational activation function? The definition says $R(x)$ where x is a real number, but the equation uses $x^j$, what is $x^j$?

------------------
I've updated my score in light of the new results.

---

> ### Author Response · Authors · 2023-11-22
>
> Dear Reviewer,
>
> we thank you for your valuable feedback and the novel work you have brought to our attention. The points raised have help us to greatly improve the manuscript. Let us address the different points that you have raised. Please find the updated manuscript, with all the modifications highlighted in blue.
>
> **Missing literature on plasticity.** Sorry for this. We added an entire paragraph called **_Plasticity in deep RL_** in our related work section, discussing all the work you have provided us with. In our Fig. 1, we have now added results of agents that use CReLU and commented on it.
>
> **Better performance does not mean better plasticity.** Rational activation functions (compared to rigid baseline) increase the adaptability of the network, as they learn to reshape the weights’ optimization landscape. This is also shown by the fact that they reduce with overestimation, as the authors of DDQN (developed to tackle overestimation), explain and show that improved plasticity reduces overestimation [1]. However, to make this point clearer, we agree that the permutted MNIST experiment would show our point. We thus added it in our manuscript (in Appendix A1) and refer to it in 2.1. We trained ReLU, CReLU and rational networks sequentially on MNIST, a first permuted version of it, and a second permuted one. Results show that networks equipped with rational activation functions are able to better model the new incoming data, while retaining more of the already fitted one.
>
> **Question**:
>
> x^j in the equation corresponds to “x power j” (j is the exponent, not an index). Rationals are ratios of polynomials. We clarified that i and j are exponents in our manuscript to avoid any confusion, thank you.
>
> [1] Hasselt, H. V. et al. “Deep Reinforcement Learning with Double Q-Learning.” _AAAI Conference on Artificial Intelligence_ (2015).

---

> ### Comment · Reviewer_7Gju · 2023-11-23
> **Updated score to recommend acceptance**
>
> Dear Authors, thank you for your response. The new results are interesting, and they alleviate my original concerns. The experiments in the supervised learning problem show that rational activations do help with plasticity, which significantly improves the strength of the claims in the paper. In light of these results, I have increased my score.
> I recommend the area and program chairs accept this paper as other reviewers have not responded to the rebuttal. But, the updated paper also addresses most of the concerns other reviewers raised.

---

> > ### Author Response · Authors · 2023-11-23
> > **Thank you very much for your implication.**
> >
> > Dear reviewer, we really appreciate the time you are taking for considering our updated manuscript.
> > We are happy that we addressed your original concerns.
> > Thank you very much for your implication!

---

### Official Review · Reviewer_cKgE · 2023-11-02

**Soundness:** 2 fair
**Presentation:** 3 good
**Contribution:** 3 good
**Rating:** 8
**Confidence:** 4

**Summary:**

The paper argues for plasticity in RL through adaptive activation functions. It proposes using (safe) rational activation functions as a plug-in modification to existing value-based RL methods (DQN-style algorithms).

The authors first motivate the need for plasticity through activation functions in constantly changing reinforcement learning environments. They categorize environments into static, dynamic, and progressively evolving environments. The authors then show how existing methods, such as DQN and non-rational plasticity variants (PELU), fail to adapt to all changes, whereas rational activations can.

Having established Rational activation functions as promising candidates for imbuing existing neural networks with plasticity, the authors show that rational activation functions can embed residual connections. Using this insight, they develop a joint rational activation function that is a regularized form of rational plasticity through weight-sharing across activations in different layers.

The authors finally demonstrate through experiments that equipping popular algorithms such as a DQN with rational activations leads to consistent improvements in performance on Atari games, elevating them to be competitive alternatives to highly engineered and computationally expensive methods such as RAINBOW.

**Strengths:**

### Originality
The proposed application of the activation function to the RL pipeline is a novel contribution.

### Quality
The quality of the work is generally high.

### Clarity
The work is generally presented clearly.

### Significance
Activation function optimization has shown considerable success in other areas, such as pruning. Therefore, applying them as an alternative to heavy architectural search and per-environment hyperparameter optimization methods is a significant step.

**Weaknesses:**

- The work seems to miss some previous work on plasticity in RL. Please see the questions section for more information on related works. Overall, I think discussing this would further concretize the argumentation for rational plasticity while placing this work better in the RL literature.
- Hyperparameters are not at all discussed as a requirement for adaptation. I believe this is a critical aspect of designing RL pipelines, and therefore, a focus on only adaptable Neural architectures is limiting, in my opinion. It would make a lot of sense if the authors explicitly state how they see their work as related to this literature.
- I think the baselines need more justification. While it is understandable that the focus is on activation functions, so comparisons with static methods make sense, the authors also argue that the dynamically changing nature of rational activations allows RL agents to adapt to changing environmental circumstances. This usability argument, in my opinion, is not captured by the considered baselines since, most of the time, when we are talking about addressing changing aspects of environments, the focus is either on algorithmic improvements, continual and curriculum-based methods, or dynamic hyperparameter adaptation mechanisms. I believe either making strong arguments as to why the authors do not consider these approaches or performing comparisons to one of them would significantly boost this line of argumentation.

**Questions:**

- How is this rational plasticity related to previous works on plasticity in RL? [Sokar et. al, 2023, Abbas et. al, 2023,Lyle et al., 2023,Nishikin et al., 2022]
- Parametric forms to capture existing activation functions have previously been used in supervised learning [Loni et al., 2023]. How is the rational form placed regarding these works? Does the comparison to SiLU and PELU capture this already?
- It is argued in sections 1 and 2 that RL can significantly benefit from adaptive neural architectures. However, another critical aspect of RL algorithms is hyperparameter optimization, with potentially the requirement to adapt hyperparameters as an RL agent dynamically trains [Mohan et al., 2023]. To what extent can the adaptable rational activation functions in RL mitigate these requirements?
- Since the benefit of applying rational activation functions lies in their ability to adjust their parameters dynamically, shouldn't they be additionally compared to Baselines of dynamic hyperparameter optimization, such as hyperparameter schedules [Zahavy et al., 2020], progressive episode lengths [Fuks et al., 2019]? I think it makes sense to at least justify whether relationships between the use of rational activation functions for improving adaptability and methods to tackle the same issue in this literature does exist or not.

### References
- [Sokar et. al, 2023] Ghada Sokar, Rishabh Agarwal, Pablo Samuel Castro, and Utku Evci.  The dormant neuron phenomenon in deep reinforcement learning. arXiv preprint arXiv:2302.12902, 2023.
- [Abbas et. al, 2023] Zaheer Abbas, Rosie Zhao, Joseph Modayil, Adam White, and Marlos C Machado. Loss of plasticity in continual deep reinforcement learning. arXiv preprint arXiv:2303.07507, 2023.
- [Lyle et al., 2023] Clare Lyle, Zeyu Zheng, Evgenii Nikishin, Bernardo Avila Pires, Razvan Pascanu, and Will Dabney. Understanding plasticity in neural networks. arXiv preprint arXiv:2303.01486, 2023.
- [Nishikin et al., 2022] Evgenii Nikishin, Max Schwarzer, Pierluca D’Oro, Pierre-Luc Bacon, and Aaron Courville.  The primacy bias in deep reinforcement learning. In International Conference on Machine Learning, pages 16828–16847. PMLR, 2022.
- [Loni et al, 2023] Loni, M., Mohan, A., Asadi, M., & Lindauer, M. (2023). Learning Activation Functions for Sparse Neural Networks. arXiv preprint arXiv:2305.10964.
- [Mohan et al, 2023] Mohan, A., Benjamins, C., Wienecke, K., Dockhorn, A., & Lindauer, M. (2023). Autorl hyperparameter landscapes. AutoML Conference 2023 (https://openreview.net/forum?id=Ec09TcV_HKq).
- [Zahavy et al., 2020] Zahavy, T., Xu, Z., Veeriah, V., Hessel, M., Oh, J., van Hasselt, H. P., ... & Singh, S. (2020). A self-tuning actor-critic algorithm. Advances in neural information processing systems, 33, 20913-20924.
- [Fuks et al., 2019] Fuks, L., Awad, N. H., Hutter, F., & Lindauer, M. (2019, August). An Evolution Strategy with Progressive Episode Lengths for Playing Games. In IJCAI (pp. 1234-1240).

I have increased my score based on the changes made by the authors

---

> ### Author Response · Authors · 2023-11-22
> **Thank you for all your work. Please consider our answer.**
>
> Dear Reviewer,
>
> we really want to thank you for all the effort you have put in the reviews, and the valuable feedback you have provided us, notably all the related work that we were able to insert in our manuscript. Your raised points have greatly helped us to improve our manuscript. Let us address the opportunities (which we enumerated to make the assignment easier). The updated manuscript has the modifications done in blue.
>
> **Missing related work.** Sorry for this, and thank you so much for all these references. We studied these works and now discuss them in the new **_Plasticity in RL_** paragraph of our Related Work section (in blue in the updated manuscript). In short, apart from CReLU, (that we now compare to in both RL and CL experiments, cf. Fig. 2 and 6), all techniques could be applied together with rational plasticity.
>
> **Discuss links with hyperparameter adaptation**. We agree, and have added a discussion both in the **_Related Work_** and in the **_Limitation and Future Work_** sections. As the rational functions can reshape the weights' optimization landscape, some hyperparameter adaptation might already be covered. It seemed to us that rationals automatically adapted this landscape, testified by shape changes, both through time (showed by Fig. 1) and among environments complexities (c.f. Fig. 9 in Appendix). Testing it further be very interesting to test, particularly in continual settings, where noise would be reduced.
>
> **Why not other Continual and curriculum learning methods ?** We believe that the approaches relevant to these subfields and our rational plasticity are complementary, as many complementary techniques need to be combined for continual learning [1]. Combining these different techniques, both in RL and CL, discussing the benefits and drawbacks of each and their complementarity would be a great benefit for our community. We plan on integrating rational plasticity into more continual learning experiments. However, it cannot fit in this work without making it too dense, and is thus left for future works.
>
>
>
> **Questions:**
>
> **Parametric activations have already been used, how to place rationals in comparison to them?** Indeed, parametric versions of classic rigid activation functions have been developed and used (e.g. PReLU), as well as linear combinations of existing functions. We discuss this in the introduction and (paragraphs 2 and 3) and in the **“The choice of activation functions”** paragraph of our related work sections. In short, rational functions being universal approximants, they have higher expressivity than linear combinations, and with meaningful choices of the degrees (also discussed), present better stability and approximation capabilities than rationals (see also [2])
>
> **Comments on rational activation functions and methods to improve adaptability.**
>
> Rational functions can dynamically adjust their slopes (and thus the gradient that pass through them). This is notably the case for the functions that evolve on the TimePilot environment (depicted in Fig. 1), both layer-wise and epoch wise. We comment that even further in the newly added Fig. 10 in the appendix, about our permitted MNIST experiment.
>
> [1] Kudithipudi, Dhireesha et al. “Biological underpinnings for lifelong learning machines.” _Nature Machine Intelligence_ 4 (2022).
>
> [2] Telgarsky, Matus. “Neural Networks and Rational Functions.” _ArXiv_ abs/1706.03301 (2017)

---

> > ### Comment · Reviewer_cKgE · 2023-11-23
> >
> > Dear Authors,
> >
> > Thank you very much for the changes. The paper looks much better now, with an improved related works section and the CRelu baseline. I also particularly liked the intuitive diagrams in the appendix, which make the methodological architecture more transparent and better ground the evaluation procedure.
> >
> > In light of these changes, I am increasing my score.
> >
> > Some grammatical and formatting errors in the newly added text:
> > - Section 2.1:  Permutted-MNIST --> Permuted-MNIST
> > - Section 3 , Q3: van Hasselt et al. --> formatting looks off
> > - Section 4: is also improves --> also improves, weights--> weights'
> > - A.1 plasticy --> plasticity
> > - A.8: evolutions  --> evolution
> > - A.9: halves --> halved

---

### Meta-Review · Area_Chair_GkFQ · 2023-12-09

**Metareview:**

This paper proposes a safe, rational activation function as a plug-in modification to existing value-based RL methods. Experimental evaluation on Atari games reveals substantial gains. There were concerns about the clarity of the paper and the theorems in the paper. After several discussions, all reviewers have rated this work positively.

**Justification For Why Not Higher Score:**

There were concerns of clarity along with novelty of proposed methods vis-a-viz prior work in neural plasticity.

**Justification For Why Not Lower Score:**

Paper is overall quite solid and significantly improves upon prior work.

---

### Decision · Program_Chairs · 2024-01-16

Accept (spotlight)